# Evaluation of New CORDEX Simulations Using an Updated Köppen–Trewartha Climate Classification

**Armelle Reca Remedio *** **, Claas Teichmann** , **Lars Buntemeyer** , **Kevin Sieck** ,
**Torsten Weber** , **Diana Rechid, Peter Hoffmann** , **Christine Nam** , **Lola Kotova** **and**
**Daniela Jacob**

Climate Service Center Germany (GERICS), Helmholtz-Zentrum Geesthacht, 20095 Hamburg, Germany;
claas.teichmann@hzg.de (C.T.); lars.buntemeyer@hzg.de (L.B.); kevin.sieck@hzg.de (K.S.);
torsten.weber@hzg.de (T.W.); diana.rechid@hzg.de (D.R.); peter.hoffmann@hzg.de (P.H.);
christine.nam@hzg.de (C.N.); lola.kotova@hzg.de (L.K.); daniela.jacob@hzg.de (D.J.)
* Correspondence: armelle.remedio@hzg.de; Tel.: +49-4022-6338-451

**Abstract:** A new ensemble of climate and climate change simulations covering all major inhabited regions with a spatial resolution of about 25 km, from the WCRP CORDEX COmmon Regional Experiment (CORE) Framework, has been established in support of the growing demands for climate services. The main objective of this study is to assess the quality of the simulated climate and its fitness for climate change projections by REMO (REMO2015), a regional climate model of Climate Service Center Germany (GERICS) and one of the RCMs used in the CORDEX-CORE Framework. The CORDEX-CORE REMO2015 simulations were driven by the ECMWF ERA-Interim reanalysis and the simulations were evaluated in terms of biases and skill scores over ten CORDEX Domains against the Climatic Research Unit (CRU) TS version 4.02, from 1981 to 2010, according to the regions defined by the Köppen–Trewartha (K–T) Climate Classification types. The REMO simulations have a relatively low mean annual temperature bias (about ±0.5 K) with low spatial standard deviation (about ±1.5 K) in the European, African, North and Central American, and Southeast Asian domains. The relative mean annual precipitation biases of REMO are below ±50% in most domains; however, spatial standard deviation varies from ±30% to ±200%. The REMO results simulated most climate types relatively well with lowest biases and highest skill score found in the boreal, temperate, and subtropical regions. In dry and polar regions, the REMO results simulated a relatively high annual biases of precipitation and temperature and low skill. Biases were traced to: missing or misrepresented processes, observational uncertainty, and uncertainties due to input boundary forcing.

**Keywords:** climate classification; CORDEX; REMO; CRU; model biases; observational uncertainty; input boundary forcing; ERA-Interim reanalysis

## 1. Introduction

Over the last few decades, regional climate models are increasingly used as a tool for understanding regional scale phenomena and assessing possible future climate change impacts. The demand for an ensemble of climate simulations at regional levels has resulted in initiatives such as the World Climate Research Program (WCRP) Initiative on COordinated Regional Downscaling EXperiments or CORDEX [1]. With the growing demand for high-resolution information about regional climate change and its impacts all over the world, the WCRP CORDEX is supporting the CORDEX-COmmon Regional Experiment (CORE) Framework [2]. CORDEX-CORE aims to contribute to the next Intergovernmental Panel on Climate Change (IPCC) report with a homogeneous dataset of high-resolution regional climate information of at least 25 km spatial resolution for all

major inhabited areas of the world. This framework has produced a baseline set of homogeneous high-resolution dynamically downscaled reanalysis forced by the ECMWF ERA-Interim [3] and projections forced by selected global climate models using the low- (rcp2.6) and high-end (rcp8.5) representative concentration pathways (rcp) scenarios [4]. With this new framework, a new ensemble of climate simulations from at least two participating regional climate centers have been created to provide as a basis for assessments of climate change scenarios as well as possible future extreme events for all major inhabited regions of the world. The simulations will be used to support the growing demands for climate services to provide scientifically sound decisions on climate change adaptation. The coordinated high-resolution simulations could also be used as a basis for further research on climate vulnerability, impacts, and adaptation.

In this study, we applied the regional climate model REMO and analyzed the results of the present climate driven by the ERA-Interim reanalysis. The REMO model was originally developed for Europe and extended to several regions over the globe such as Africa [5,6], South America [7], South Asia [8], and North America [9]. Currently, REMO is being used to study impacts of climate change over these domains. For example, REMO simulations contributed to CORDEX studies highlighting the impacts of climate change at +1.5 °C from its ensemble of climate change simulations from a resolution of 0.44° (about 50 km) over selected regions of the globe e.g., Africa [10] and down to 0.11° (about 12.5 km) over Europe [11].

Within CORDEX-CORE [12], new high resolution simulations of 0.22° (about 25 km) were set up over most domains, except for Southeast Asia [13,14] and Europe [15], which were already at this resolution or higher (at about 12.5 km over Europe). These new high resolution simulations will provide additional climate simulations over regions especially with few ensemble members e.g., over Central Asia [16] and Central America [17]. Figure 1 shows ten out of the fourteen CORDEX domains that were used in this study.

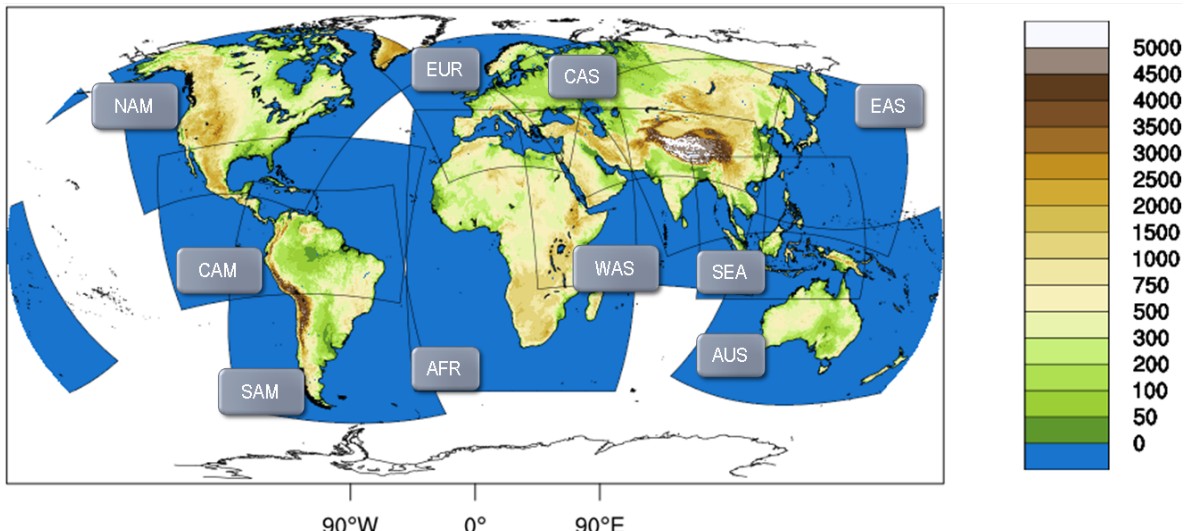

**Figure 1.** The ten CORDEX-CORE model domains simulated by the model REMO: North America (**NAM**), Central America (**CAM**), South America (**SAM**), Europe (**EUR**), Africa (**AFR**), South Asia (formerly referred to as West Asia, **WAS**), Central Asia (**CAS**), East Asia (**EAS**), Southeast Asia (**SEA**), and Australasia (**AUS**). The orography is shown in m. The domain boundaries are indicated with black polygons.

The regions of the world can be subdivided into political boundaries or into different climate types based on its long-term precipitation and temperature characteristics. Defining the regions of analysis according to climate types, rather than political zones, provide regional information based on physical processes. Using threshold values of temperature and precipitation, the global climate was originally classified by Köppen [18]. A modification of these thresholds and using an updated

observational datasets produced classification schemes such as the Köppen–Trewartha (K–T) Climate Classification [19,20]. The similarities of the latter to the classical scheme were thoroughly discussed in Belda et al. [21]. The K–T climate types have been widely used in previous studies [22–25] due to its similarities with the native vegetation. In this study, the K–T climate types were used to define the regions of analysis using an updated observational dataset.

Before estimating the climate change signal, an evaluation of the model performance for each domain is needed in order to provide estimates of uncertainties from the model simulations. The objective of this study is to investigate temperature and precipitation biases over the new high resolved domains, as well as the skill of the model in representing the climatology of all regions in order to identify possible sources of systematic errors of the model inherent to input boundary forcing, observational uncertainty (e.g., [26]), or misrepresented processes (e.g., [27]). This work builds on previous studies which evaluated the skill of representing the climate over individual domains (e.g. Europe [28], Africa [29], South America [30]) using multi-models ensembles. The skill of the model is quantified by using probability density functions (PDF) of the observed and simulated temperature and precipitation aggregated at each climate type and each region following the PDF skill score method [31]. This method provides a robust comparison of the similarity between the PDF of the simulated and observed values.

The REMO model and experimental setup as well as the regions of analysis using the K–T climate types are discussed in Section 2. Section 3 shows the climate statistics and results of the model evaluation including the biases and skill of temperature and precipitation. Section 4 discusses the sources of the systematic errors in the model relating this errors to the mean annual cycles followed by conclusions in Section 5.

## 2. Data and Methods

### 2.1. REMO

The regional climate model REMO [23,32,33] is a three-dimensional, hydrostatic, atmospheric circulation model within a limited area. REMO is based on the physical paramaterizations of ECHAM 4.5 [34] and the dynamical core of the German Weather Service (DWD) Europa–Modell (EM) weather prediction model [35]. The REMO model is used for various studies on coupling with other earth system components such as with aerosols (REMO-HAM)[36], lakes (REMO-FLAKE) [37], interactive mosaic-based vegetation (REMO-iMOVE) [38], and global ocean model (ROM) [39]. Most of these model developments have been performed over the European domain.

In this study, the latest hydrostatic version of REMO (REMO2015) is used with 27 hybrid sigma-pressure coordinate system, which follows the surface orography in the lower levels but independent from it at higher atmospheric model levels. REMO2015 has a leap-frog time stepping with semi-implicit correction and Asselin-filter. The prognostic variables are surface pressure, temperature, horizontal wind components, water vapor, and cloud water content.

The mass flux convection scheme is parameterized after Tiedtke [40] with modifications after Nordeng [41] and Pfeifer [42]. The stratiform cloud scheme calculates prognostic equations for the vapor, liquid, and ice phase, and a cloud microphysical scheme by Lohmann and Roeckner [43]. The radiation scheme is after Morcrette et al. [44] with additional greenhouse gases, the 14.6 μm band of ozone, and various types of aerosols [45]. The aerosols are based on the Tanre [46] climatology [36]. Turbulence vertical diffusion is based on Louis [47]. The land surface scheme is based on the surface runoff scheme [48], inland glaciers [49], and vegetation phenology [50,51]. The soil hydrology is represented by a bucket scheme, in which a bucket exists at each land point and the depth of the bucket is equivalent to the mean rooting depth of the grid box [49].

In this study, the REMO model was used to simulate the ten domains shown in Figure 1. The other model parameters such as number of grid boxes (*x* and *y*), minimum cloud height over land (ZDLAND-L), or ocean (ZDLAND-O) for which rain can fall, and the Charnock constant (CHAR) for



each domain are listed in Table 1. In this setup, the largest domains were Africa (AFR), East Asia (EAS), and Australasia (AUS) while the smallest domains were Europe (EUR) and Southeast Asia (SEA). Since ZDLAND is latitude-dependent, different values were set for domains covering the tropics and the domains covering the mid-latitudes. CHAR is also a parameter that can vary in different regions due to its dependency of the general characteristics of the sea conditions.

The REMO standard value for ZDLAND, which was originally developed over Europe, is 750 m $\cdot$ g, where g is the acceleration due to gravity and assuming a cloud height of 750 m. The tropical value usually reaches the height of 1500 m or 3000 m. In tropical domains (e.g., WAS, CAM, CAS, EAS, and SEA), the minimum cloud height over land and ocean before precipitation occurs is higher compared to the other domains. The Charnock constant (CHAR), which determines the strength of the relationship between the low level winds (i.e., friction velocity) and the roughness-length over sea within the Charnock formula [52], has the same values except on WAS based on sensitivity studies on this domain [53]. The parameters for each model domain are listed in Table 1. Although the EURO-CORDEX [15] community already have simulations with a very high spatial resolution of 12.5 km (EUR-11), in this study, we run EUR-22 simulation as a benchmark for the simulations.

**Table 1.** Model parameters used in the ten domains (Figure 1): Europe (**EUR**), Africa (**AFR**), South Asia (**WAS**), North America (**NAM**), South America (**SAM**), Central America (**CAM**), Central Asia (**CAS**), East Asia (**EAS**), Southeast Asia (**SEA**), and Australasia (**AUS**). The model parameters listed below are number of grid boxes ($x$ and $y$), minimum cloud height over land (ZDLAND-L) and over ocean (ZDLAND-O), and Charnock constant (CHAR).

| Parameters | EUR | AFR | WAS | NAM | SAM | CAM | CAS | EAS | SEA | AUS |
|---|---|---|---|---|---|---|---|---|---|---|
| $x$ | 241 | 401 | 401 | 321 | 301 | 433 | 325 | 433 | 288 | 433 |
| $y$ | 217 | 433 | 271 | 271 | 361 | 241 | 217 | 271 | 217 | 271 |
| ZDLAND-L | 7500 | 20,000 | 25,000 | 7500 | 7500 | 25,000 | 25,000 | 25,000 | 25,000 | 7500 |
| ZDLAND-O | 7500 | 15,000 | 30,000 | 7500 | 7500 | 30,000 | 30,000 | 30,000 | 30,000 | 7500 |
| CHAR | 0.0123 | 0.0123 | 0.00123 | 0.0123 | 0.0123 | 0.0123 | 0.0123 | 0.0123 | 0.0123 | 0.0123 |

The setup for each model domain in Table 1 were based on the best expert knowledge for the respective regions. For example, due to sensitivity studies in South America and in the South Asia domains, some surface parameters were modified. In South America, it was found that the warm bias in forested regions were reduced due to reduction of surface evaporation when the wilting point parameter was lowered in forested regions compared to non-forested regions [54]. In the South Asia domain, the soil heat conductivity was reduced to represent the dry soils, which improved the representation of climate especially over India [53].

The global reanalysis data of ERA-Interim [3], which has a horizontal resolution of about $0.7° \times 0.7°$ was used as the initial and boundary conditions of REMO. ERA-Interim was interpolated to all the ten model domains (Figure 1) from 1979 to 2017. The model was integrated with a time step of 120 s. For the model to reach an equilibrium state, a spin-up period of thirty years was implemented. In each domain, the model was initially spun-up for 30 years from 1979 to 2008 to account for the time the model needed to produce an equilibrium for the soil temperature and soil moisture. These soil fields were then used as the initial soil conditions upon restarting the model from the year 1979.

The forcing data were prescribed at the lateral boundaries of each domain, which mainly influenced the eight outer grid boxes, with an exponential decrease towards the center of the model domain using a relaxation scheme [55]. The lateral boundary conditions were updated every 6 h. As mentioned in the CORDEX-CORE experimental guidelines [12], the downscaling was conducted to a horizontal resolution of $0.22° \times 0.22°$ (approx. $25 \times 25$ km$^2$). The climate variables used in this study were precipitation and the near surface temperature or temperature at 2 m height. In this study, we refer to the near surface temperature as temperature throughout the manuscript.

## 2.2. Köppen–Trewartha Climate Classification

The analysis regions considered in this study were defined using the Köppen–Trewartha (K–T) Climate Classification [20]. The fourteen climate types (Table 2) were derived using the mean annual temperature (*Tann*, in °C) and the mean annual precipitation (*Pann*, in cm) of a 30-year climatology from an observational dataset. For climatology, we used the Climatic Research Unit (CRU) temperature and precipitation version CRU TS 4.02 dataset [56]. This version utilized a revised interpolation function improving discontinuities in regions with sparse observations for the period 1901–2017. The spatial coverage included all land areas excluding Antarctica at 0.5° resolution. The climate period considered in this study was from 1981 to 2010.

**Table 2.** Climate types defined by the Köppen–Trewartha Climate Classification [57]. The temperature metrics (°C) were the mean annual temperature (*Tann*), mean monthly temperature (*Tmon*), coldest (warmest) month *Tcold* (*Twarm*). The precipitation metrics were the mean annual precipitation (*Pann*, in cm), number of dry months (*Pdry*) or wet months (*Pwet*). The threshold for a dry (wet) month was the mean monthly precipitation $< (\geq)$ 6 cm. To distinguish the dry season, we used the Patton's precipitation threshold [21] and defined as $R = 2.3 \cdot Tann - 0.64 \cdot Pwinter + 41$, where *Pwinter* (%) was the percentage of annual precipitation occurring in winter or the lowest sunshine duration in the northern (southern) hemisphere during October to March (April to September).

| Types | Description |
|---|---|
| | A: Tropical climates; $Tcold > 18\,°C$; $Pann \geq R$ |
| Ar | humid; 10 to 12 months wet; 0 to 2 months dry |
| Aw | Winter (low-sun period) dry; $> 2$ months dry |
| As | Summer (high-sun period) dry; rare climate type |
| | B: Dry climates; $Pann < R$ |
| BS | semi-arid; $R/2 < Pann < R$ |
| BW | arid; $Pann < R/2$ |
| | C: Subtropical climates; $Tcold < 18\,°C$ |
| Cs | summer-dry; at least $3\times$ as much rain in winter as in summer; $Pdry < 3$ cm; $Pann < 89$ cm |
| Cw | summer-wet; winter dry; at least 10x as much rain in summer as in winter |
| Cf | humid; No dry season; difference between driest and wettest month less than required for Cs and Cw; $Pdry > 3$ cm |
| | D: Temperate climates; 4 to 7 months with $Tmo > 10\,°C$ |
| Do | oceanic; $Tcold > 0\,°C$ |
| Dc | continental; $Tcold \leq 0\,°C$ |
| | E: Sub-arctic or boreal climates; |
| | 1 to 3 months with $Tmo > 10\,°C$ |
| Eo | oceanic; $Tcold > -10\,°C$ |
| Ec | continental; $Tcold \leq -10\,°C$ |
| | F: Polar climates; $Twarm < 10\,°C$ |
| Ft | Tundra/Highland; $Twarm > 0\,°C$ |
| Fi | Ice cap; $Twarm \leq 0\,°C$ |

In order to make sure that each grid box have exclusive climate types, the order of obtaining climate types is the following: the polar and boreal climates (Types F and E), followed by the dry climate (Type B), the temperate climate (Type D), and then subtropical and tropical climates (Types C and A). An additional check after the run-through of the classification is the regions with an absence of precipitation (*Pann* = 0) based on the CRU dataset and with *Tcold* > 18 °C, which are then classified as an arid climate (BW).

### 2.3. Climate Statistics, Biases and Skill

The observed climate statistics were calculated such as the mean, as well as temporal and spatial variability. The model biases and skill were mainly evaluated against the CRU observational dataset. The simulations were interpolated to the CRU grid (0.5° resolution) using conservative and bilinear remapping functions from the Climate Data Operators version 1.9.7 [58]. The absolute biases of the simulated monthly precipitation and temperature were calculated compared to the observed monthly values. For the relative bias, the mean absolute bias was normalized with the observed climatological mean. The mean absolute and relative biases were then aggregated into regions with similar climate types defined in Table 2. The simulated temperature values were height-corrected to account the differences in the orography between REMO and CRU.

In order to gauge the observational uncertainty, additional global datasets were used in analyzing the mean annual cycle of temperature and precipitation: the Global Precipitation Climatology Centre (GPCC, monthly, 0.25°, [59]); the University of Delaware Temperature and Precipitation (UDEL, monthly, 0.5°, [60,61]); and the Global Land Data Assimilation System (GLDASD, daily, 0.25°, [62]). The temperature derived from ERA-Interim was also included to compare the temperature from the input boundary forcing. In the comparisons of mean annual cycles with observational datasets, we additionally compared the simulations in the CORDEX Framework [23]: Europe at 50 km (EUR-44) and 12.5 km (EUR-11); Africa at 50 km (AFR-44); North America at 50 km (NAM-44); South America at 50 km (SAM-44); and South Asia at 50 km (WAS-44). Except for EUR-11, the CORDEX simulations were done with a previous version of REMO, which is REMO2009. The main difference of REMO2015 from the previous version (REMO2009) is the additional option for a non-hydrostatic simulation, which was opted out in this study.

The skill score was based on the empirical probability functions [31] of precipitation and temperature aggregated at each climate region. The dimensionless skill score had a value between 0 to 1, with the value of 1 indicating that the probability density functions of the observed and modelled values are the same. In this study, instead of using the entire temperature and precipitation distribution [23], we measured the model skill for each season:

$$Skill_{seasons} = \sum_{i=1}^{n} \min(Zseas_m, Zseas_o),\qquad(1)$$

where $n$ is the number of bins used to calculate the normalized PDF for a given region, and $Zseas_m$ and $Zseas_o$ are the seasonal frequency of model values and observed values in a given bin, respectively. The bin sizes used were 0.1 °C for temperature and 1 mm/day for precipitation.

### 3. Results

The results of this work are divided into four subsections. The observed and simulated temperature and precipitation across the domains are presented during the period of study (Section 3.1). In Section 3.2, the results for the updated K–T climate types derived from CRU TS4.02 observational dataset are presented. This is followed by an evaluation of the regions with similar climate types, in which the regions are considered to be significant if the area is more than 5% of the domain area. The mean annual biases and model skill across the ten domains are presented in Sections 3.3 and 3.4. The detailed discussions on the causes of these biases are elaborated in Section 4.

### 3.1. Temperature and Precipitation from Observations and Simulations

In order to investigate the new ensemble of high spatial resolution simulations of 25 km, we compared the simulations with the CRU observational dataset. The observed climatology of the mean annual temperature and precipitation over the globe except Antarctica are shown in Figure 2. The temperature values vary from 24 to more than 30 °C and the precipitation values range from 6 to

20 mm/day along the low latitudes. The cold (below 10 °C) and dry (less than 2 mm/day) regions are also shown located at higher latitudes.

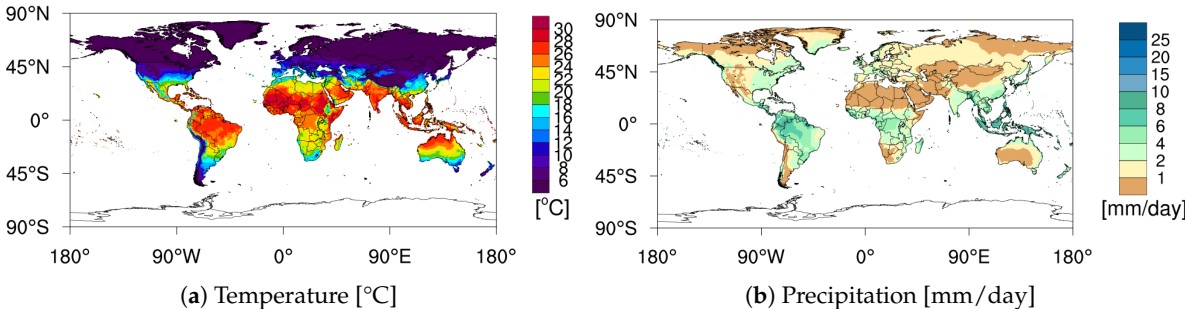

(**a**) Temperature [°C]            (**b**) Precipitation [mm/day]

**Figure 2.** Mean annual temperature (**a**) and precipitation (**b**) based on the Climatic Research Unit (CRU) TS 4.02 observational dataset during the period 1981 to 2010.

Figure 3a depicts the temperature biases compared to CRU for all the domains during the period of 1981 to 2010. The REMO simulations have a relatively low mean annual temperature bias (within the range of ±0.5 K) in EUR, AFR, NAM, CAM, and SEA domains. Regional warm and cold biases, within the range of −5 to 5 K, are found, similar to that of the previous CORDEX simulations by REMO2009, driven by the same boundary forcing, but with a coarse resolution of 50 km [23] for the EUR, NAM, AFR, SAM, and WAS domains.

There are noticeable warm biases in the coastal areas, especially Baja California in North America, Namib Desert, and Angola in Africa, and the coasts of Chile in South America. For the overlapping domains, the temperature biases were similar with varying magnitude, which will be presented later in Section 3.2. For example, the model showed a cold bias over the northern part of Africa in both the EUR and AFR domains, and it reproduced warm biases over Eastern Africa and the coasts of Yemen and Oman in both the AFR and WAS domains. The warm bias over the Amazon in the CAM domain is similarly depicted in the SAM domain.The cold bias over the island of New Guinea in the SEA domain is also present in the AUS domain. A warm bias of about 2 K can be found over the Australia continent.

In comparison to a previous CORDEX study with an older REMO model version ([23]), the cold bias over central India is reduced in the present study. The cold bias over the Himalayan region, however, still existed in this present study, and it is similarly depicted in both CAS and EAS domains, as well as in the WAS domain. The complex orography over the Himalayan region and sparse observational dataset attributed to this known modelling challenge.

Figure 3b depicts the precipitation biases compared to CRU for all the domains. The simulations produced predominantly wet biases over the European and North American continents. The dry bias in the eastern part of the African continent relate to a warm bias. A similar warm and dry bias is simulated in the model for the northeastern region of South America. In mountainous regions, such as the Himalayan mountain range and the Rocky Mountains, the model produced a wet and cold bias compared to CRU.

An interesting feature is the dry bias over the northern part of South American continent, which has a wet bias in the SAM domain. Both CAM and SAM domains have a wet bias over the Andes mountains. In coastal countries, such as Honduras and Nicaragua, however, REMO has a dry bias compared to CRU. The model has a general tendency of wet biases over the continental domains of CAS, EAS, and AUS, however, dry biases are simulated over Indonesia especially in Borneo Islands and the northwest part of the SEA domain. In the AUS domain, the underestimation of rainfall in Western Australia and the coastal regions of Australia, as well as the overestimation over the rest of Australia can be also found in other regional climate model simulations [63].

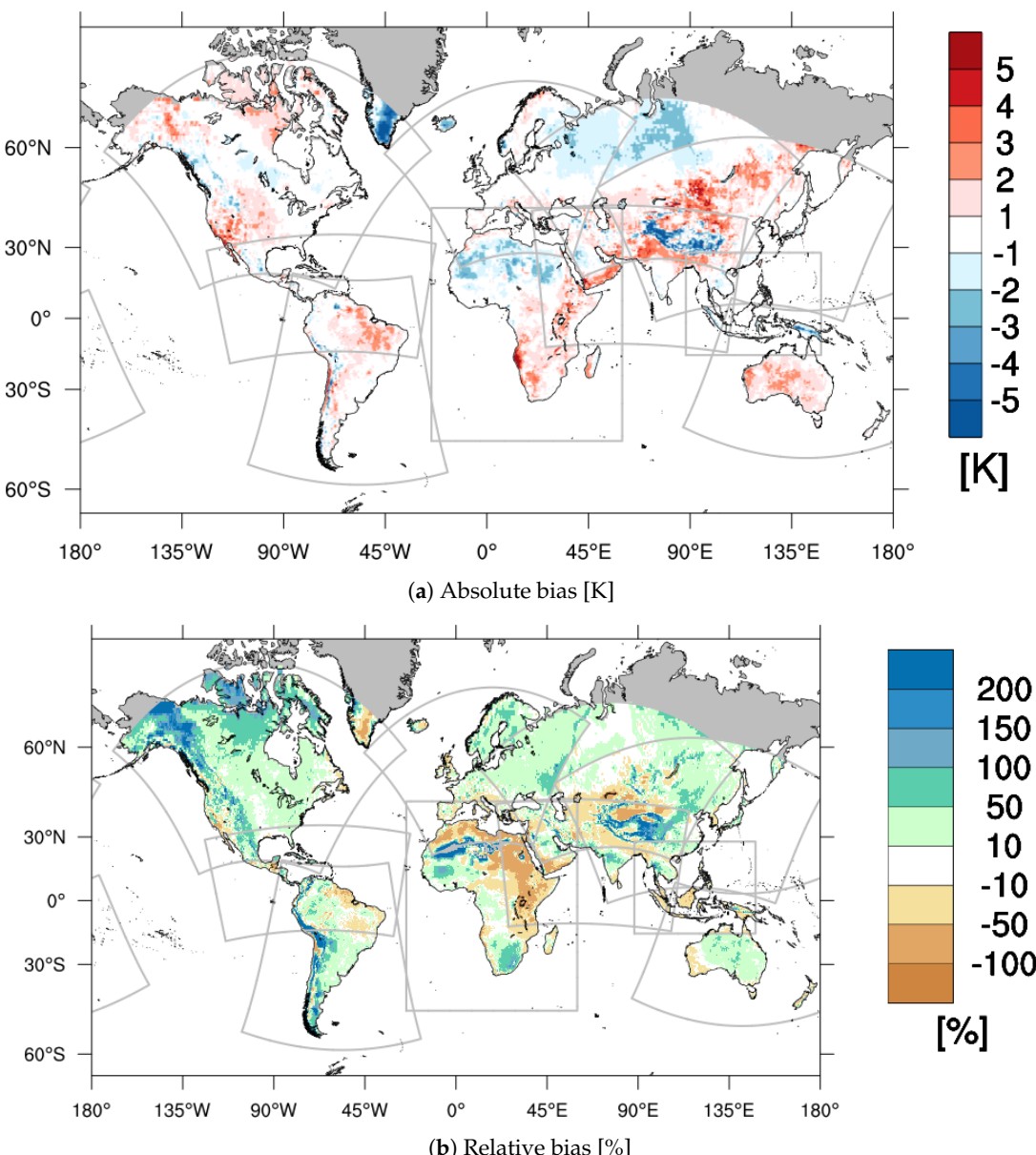

(**a**) Absolute bias [K]

(**b**) Relative bias [%]

**Figure 3.** Mean annual temperature (**a**) and precipitation (**b**) bias compared to CRU over land points only for all domains (Figure 1) during the period 1981 to 2010. The domain boundaries are indicated with gray polygons. The gray areas indicate no data. Over regions of low observed precipitation, the simulations could easily reach a relative bias of more than 100%.

Table 3 summarizes the observed temperature mean and temporal variability (temporal standard deviation or STD) as well as the biases and the spatial variability (spatial STD) of each domain. In the large domain such as CAS, EAS, and NAM, the observed variability (temporal STD) is rather high (more than ±9 K) while the mean annual temperature over the domain is low. The domain-averaged bias of REMO and the spatial STD will be used in the discussion of the biases in Section 4.

**Table 3.** Temperature statistics (mean, standard deviation or STD, and biases) for each domain (Figure 1) compared to the Climatic Research Unit (CRU) observational dataset from 1981 to 2010. The units of the CRU mean temperature are in °C while the units of the REMO bias are in K.

| Domains | CRU Mean | Temporal STD | REMO Bias | Spatial STD |
|---------|----------|--------------|-----------|-------------|
| EUR | 11.32 | ±7.79 | −0.40 | ±1.14 |
| NAM | 4.35 | ±9.80 | +0.34 | ±1.24 |
| AFR | 23.14 | ±4.20 | +0.29 | ±1.46 |
| SAM | 22.01 | ±1.94 | +0.63 | ±1.24 |
| WAS | 19.21 | ±5.46 | +0.36 | ±1.94 |
| CAM | 24.32 | ±1.79 | +0.48 | ±1.13 |
| CAS | 6.96 | ±10.72 | +0.01 | ±2.01 |
| EAS | 9.18 | ±9.69 | +0.51 | ±2.01 |
| SEA | 23.15 | ±2.80 | +0.03 | ±1.37 |
| AUS | 22.79 | ±3.49 | +0.90 | ±1.17 |

Table 4 summarizes the observed climatological precipitation mean and temporal variability (temporal STD) as well as the biases and the spatial variability (spatial STD) of each domain. The temporal variability of each domain is lower than the mean but the spatial variability of the model biases across the domains range from ±30% to almost ±200%.

**Table 4.** Precipitation statistics (mean, standard deviation or STD, and biases) for each domain (Figure 1) compared to the Climatic Research Unit (CRU) observational dataset from 1981 to 2010. The units are in mm/day except for the relative bias and relative spatial standard deviation (in %).

| Domains | CRU Mean | Temporal STD | REMO Bias | Spatial STD | Rel Bias | Rel Spatial STD |
|---------|----------|--------------|-----------|-------------|----------|-----------------|
| EUR | 1.50 | ±0.88 | +0.22 | ±0.55 | +26.11 | ±137.42 |
| NAM | 1.85 | ±1.15 | +0.50 | ±0.81 | +40.29 | ±56.21 |
| AFR | 1.62 | ±1.46 | +0.85 | ±2.47 | +15.92 | ±192.80 |
| SAM | 4.28 | ±2.81 | −0.08 | ±0.72 | +40.40 | ±110.76 |
| WAS | 1.91 | ±1.73 | +0.05 | ±1.46 | +7.43 | ±99.86 |
| CAM | 4.74 | ±3.15 | −0.78 | ±1.97 | −7.35 | ±63.07 |
| CAS | 1.25 | ±1.01 | +0.28 | ±1.24 | +23.62 | ±98.09 |
| EAS | 2.04 | ±1.79 | +0.34 | ±1.42 | +24.93 | ±101.00 |
| SEA | 5.32 | ±3.69 | −0.36 | ±2.47 | −0.30 | ±39.10 |
| AUS | 3.39 | ±2.12 | +0.64 | ±2.23 | +22.44 | ±31.35 |

### 3.2. Derived Climate Regions

The fourteen climate types derived using the definitions in Table 2 and the observed climate (Figure 2) are drawn in Figure 4, which includes the climate regions dependent on rainfall and temperature (A, B, C) and regions influenced by temperature or the thermal zones (D, E, F). The climate types based on the K–T climate classification are listed in Table 5 and the dominant climate types in the regions of the world (excluding Antarctica) relative to the total land area are: arid/desert regions (BW, 18%), sub-arctic continental (Ec, 13%), tropical winter dry (Aw, 13%), semi-arid regions (Bs, 12%), and temperate continental (Dc, 11%). The As or tropical summer dry climate zone is a rare climate type, which accounts for only about 0.1% of the total land area. The climate types that covers less than 5% of the total area in each model domain are removed in this study, which also include the rare climate types (As, Fi, Cw, and Eo). For climate change studies, the rare climate types should also be included. Note that the subtropical humid climate type (Cf) is present across the ten domains.

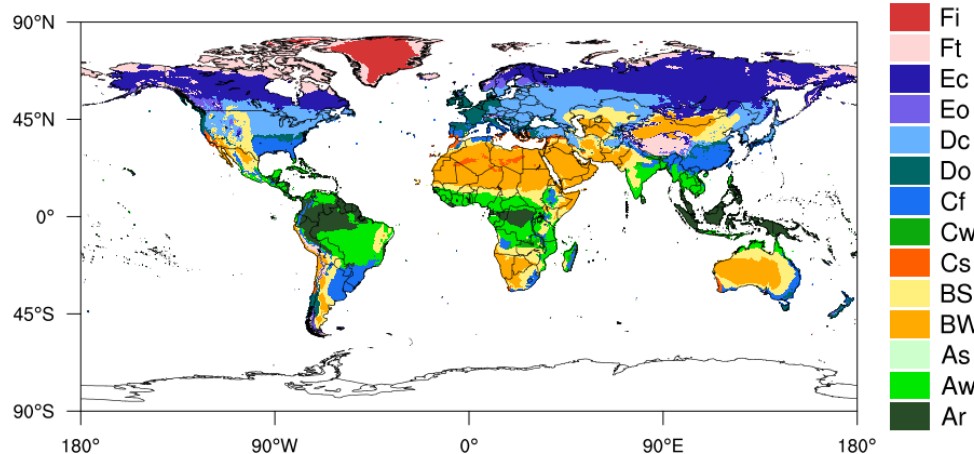

**Figure 4.** Derived global climate regions based on the Köppen–Trewartha climate classification (except Antarctica) based on CRU TS 4.02 dataset for the period 1981 to 2010.

Table 5 measures the area coverage of each climate type (in % related to the domain size). The grid area calculated is equivalent to the continental area including lakes and rivers. Of the 110 climate type regions, only 50 have an area more than 5% of its respective domain.

If we consider the land area of each 14 climate types per domain, the largest land areas are in the domains of Africa (AFR), Central Asia (CAS), South Asia (WAS), and East Asia (EAS). The domains with the smallest land area are Southeast Asia (SEA), Australasia (AUS), Central America (CAM), and Europe (EUR).

**Table 5.** The area (%) of each climate type (Table 2) relative to its respective model domain (Figure 1). The total land area for the "WORLD" (excluding Antarctica), and for each domain are depicted at the bottom row.

| Types | "WORLD" | EUR | AFR | WAS | NAM | SAM | CAM | CAS | EAS | SEA | AUS |
|---|---|---|---|---|---|---|---|---|---|---|---|
| Ar | 8.7 | - | 4.34 | 4.94 | 1.00 | 28.37 | 36.03 | - | 2.89 | 44.50 | 27.09 |
| As | 0.1 | - | 0.05 | 0.02 | - | 0.27 | 0.33 | - | 0.02 | - | - |
| Aw | 13.1 | - | 19.29 | 15.64 | 2.53 | 33.88 | 36.69 | 0.20 | 9.72 | 29.42 | 8.72 |
| BS | 12.7 | 4.74 | 16.96 | 15.42 | 10.31 | 10.21 | 10.45 | 13.71 | 14.82 | 1.31 | 21.65 |
| BW | 18.1 | 18.92 | 43.32 | 33.75 | 3.66 | 4.84 | 4.81 | 19.22 | 12.78 | - | 28.62 |
| Cf | 8.2 | 6.58 | 5.69 | 9.31 | 8.31 | 13.85 | 10.76 | 6.21 | 12.61 | 22.58 | 9.74 |
| Cs | 1.4 | 6.12 | 4.08 | 2.14 | 0.89 | 0.64 | 0.12 | 0.91 | 0.01 | - | 1.07 |
| Cw | 0.5 | - | 1.24 | 1.12 | - | 0.04 | - | 0.49 | 0.74 | 0.51 | - |
| Dc | 11.4 | 31.97 | 2.64 | 5.89 | 25.21 | 0.01 | - | 24.47 | 16.53 | 0.18 | 0.05 |
| Do | 3.5 | 15.44 | 2.29 | 3.10 | 4.65 | 2.59 | 0.04 | 3.01 | 2.46 | 1.36 | 2.73 |
| Ec | 13.4 | 10.83 | 0.07 | 1.81 | 25.15 | - | - | 24.98 | 19.18 | - | - |
| Eo | 1.6 | 4.09 | - | 0.86 | 3.67 | 1.81 | 0.08 | 0.97 | 1.26 | 0.06 | 0.30 |
| Fi | 1.0 | - | - | - | 0.77 | - | - | - | - | - | - |
| Ft | 6.2 | 1.30 | 0.02 | 6.00 | 13.85 | 3.47 | 0.70 | 5.84 | 6.98 | 0.09 | 0.02 |
| Total land area ($\times 10^6$ km$^2$) | 146 | 16 | 40 | 32 | 23 | 19 | 15 | 35 | 30 | 9 | 13 |

A few of the climate masks are shown in Figure 5. Note that the threshold used in determining the climate types is only dependent on the climatological mean of temperature and precipitation. The regions of analysis are aggregated into similar climate types where the dominant climate types such as arid (BW) or continental boreal (Ec) climate are clustered together. In some climate types e.g., BS (semi-arid dry climate), the regions are scattered since they transition from dry climates to subtropical climates. Within this transitional climate types, further analysis is recommended to consider physical processes such as the atmospheric flows. Regions influenced by the same atmospheric flow, as well as overlapping regions (e.g., BW in both WAS and CAS domains in Figures 5c,d, respectively), should be analyzed further.

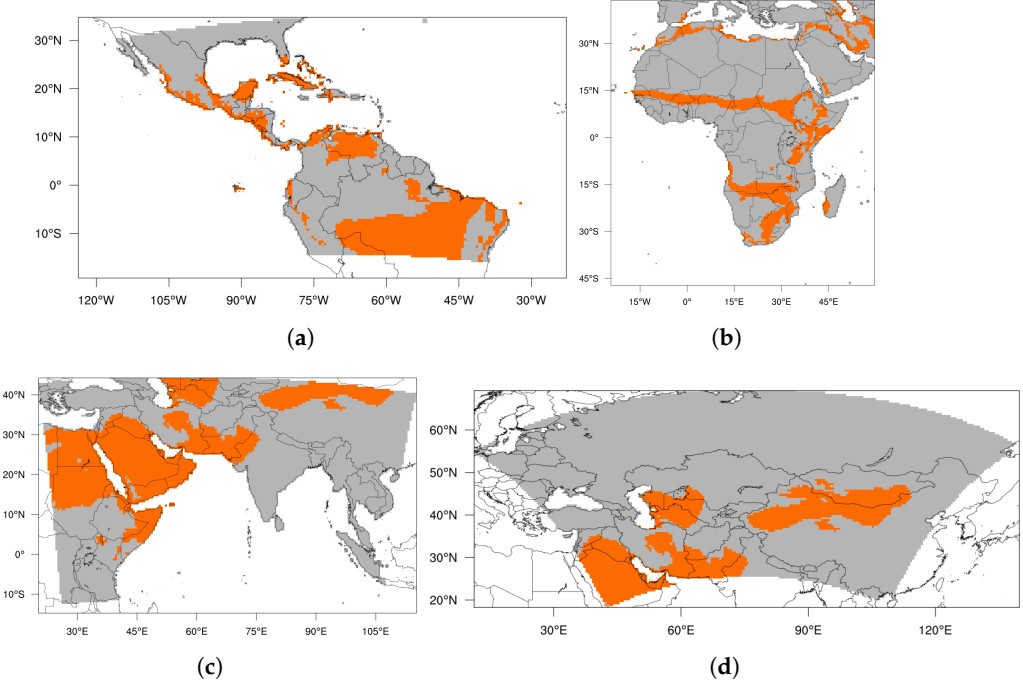

**Figure 5.** Examples of climate type regions. The orange area indicates the location of each climate type in its respective domains. (**a**) Winter dry tropical climate (Aw) in Central America (CAM); (**b**) Semi-arid dry climate (BS) in Africa (AFR); (**c**) Arid climate (BW) in South Asia (WAS); (**d**) BW in Central Asia (CAS).

### 3.3. Mean Annual Biases Using the K–T Climate Types

Based on the results from the previous section, the temperature and precipitation biases of the model were aggregated according to regions with similar climate types (Figure 4) for each domain. Figure 6 shows the mean annual bias aggregated at each climate regions in each of the ten domains. The annual bias for temperature (Figure 6a) ranges from −2.2 to 1.7 K. The bias over EUR is relatively low from −1.1 to 0.5 K. The coldest bias of about −2 K occurred in the tundra (FT) climate regions in the WAS, EAS, and CAS domains and within the observed temporal STD and spatial STD of the bias (Table 3). Similarly, the case of the warmest bias (about 1.7 K) is located in the semi-dry arid (BW) climate regions in the EAS and AUS domains. In this BW climate regions, except for EUR (cold bias, approx. −1.1 K) and AFR (low cold bias, approx. −0.2 K), the model produces a warm bias to the rest of the domains (WAS, CAS, EAS, and AUS). For arid regions (BS), all domains with this climate type have biases ranging from 0.34 K (CAM) to 1 K (WAS, AUS). In domains with similar climate types (e.g., Cf—subtropical humid climate), the model produces a warm bias in both the CAS and SEA domains, which will be a possible study on the overlapping regions.

Generally, the model tends to have warm biases especially in the tropical and subtropical climate regions (climate types A, B, C), except for the BW region in EUR (where most of the grid boxes are located in AFR) and the Ar region in SEA and AUS. In high latitudes or polar climates, the model tends to have cold biases especially in the WAS, CAS and EAS domains. For the boreal subcontinental (Ec) climate type, the model exhibits opposite biases in the CAS and EAS domains. Note that the Ec in CAS has a larger area compared to EAS (Table 5).

Figure 6b shows the relative mean annual precipitation bias aggregated at each climate region in each of the ten domains. The annual bias for precipitation ranges from about −25% to more than 170%. The annual precipitation bias over EUR ranges from −15% to 55%. The driest bias is located in the semi-arid (BW) climate regions in the WAS, EAS, and CAS domains. The wettest biases occurred in the Ft climate in the WAS, EAS, and CAS domains. A very cold bias was also depicted in Figure 6a over the Himalayas. In similar climate types (e.g., Aw—tropical winter-dry climate), the SAM and

CAM domains have opposing biases (wet bias in SAM while dry bias in CAM). Note that, in these two domains, the ZDLAND parameters differ considerably (Table 1), which might be part of the possible reasons for the differences.

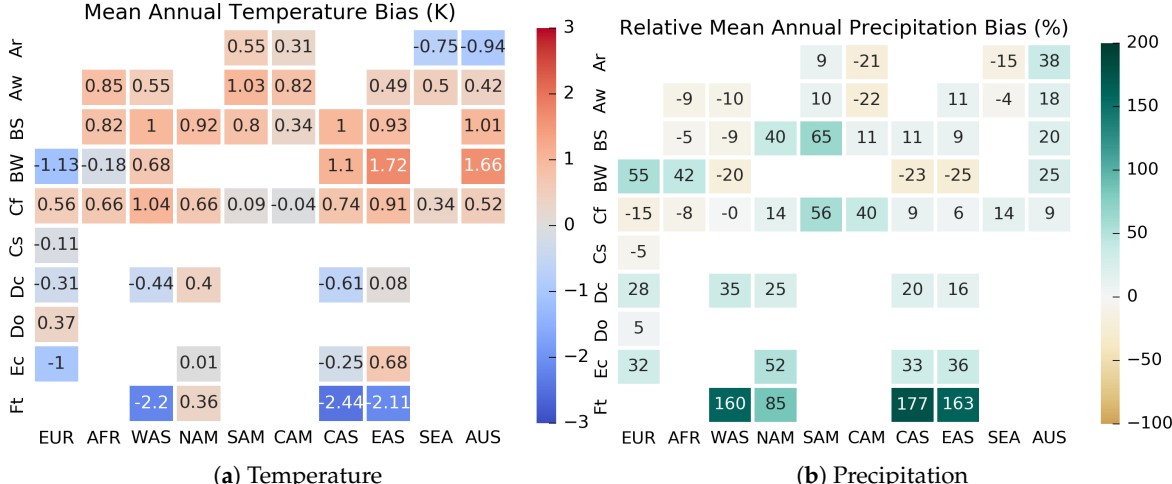

(**a**) Temperature　　　　　　　　　　　　　　　(**b**) Precipitation

**Figure 6.** Heat maps of annual mean temperature [K] and precipitation biases [%] averaged over the area of the K–T climate types located in the ten domains (Figure 1) for the period 1981 to 2010. The *x*-axis are the ten domains and the *y*-axis are the K–T climate types defined in Table 2.

*3.4. PDF Skill Score*

In evaluating the skill of the model, we compared the temporal distribution of the aggregated simulated climate variables to CRU in each climate type for each domain. The normalized PDFs of the model are compared to the observed PDFs for each seasons. For example, a high skill score of the model is shown in Figure 7. In the subtropical humid (Cf) climate type over AUS (Figure 7a), the model has a relatively high skill score of 0.97 indicating that the normalized PDFs of the simulated temperature is almost similar to the observed (Figure 7b). In contrast, in a tropical winter-dry climate region (Aw), the model has a relatively low skill score of 0.56 (figure not shown).

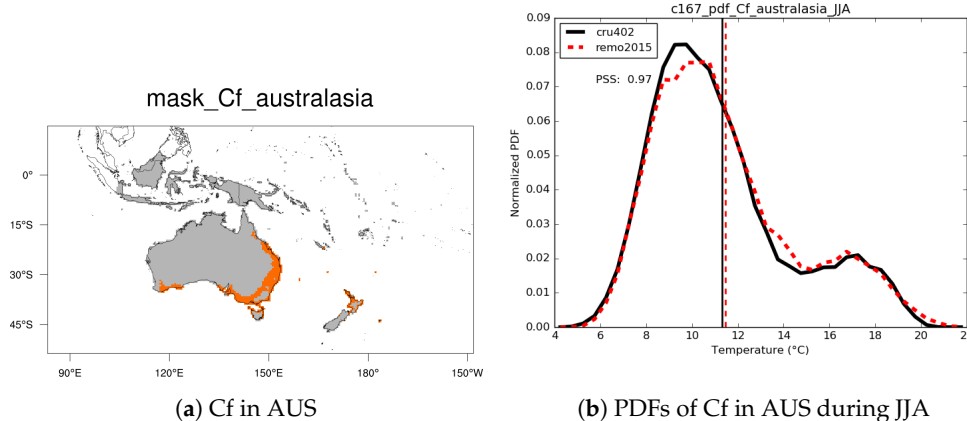

(**a**) Cf in AUS　　　　　　　　　　　(**b**) PDFs of Cf in AUS during JJA

**Figure 7.** (**a**) Sample of a climate type mask and (**b**) its normalized temperature probability density functions (PDFs) in a subtropical (Cf) climate type at Australasia (AUS) domain during June-July-August (JJA) for the period 1981 to 2010. The PDF skill score of 0.97 is indicated in the normalized PDF, and the vertical lines indicated the mean of the observed (black line) and simulated (red dashed line) distributions.

The skill scores based on the normalized PDFs for temperature and precipitation during different climatological seasons are shown in Figures 8 and 9. The skills of the model in simulating the normalized PDFs of temperature are relatively high (more than 0.8), especially during March-April-May

(MAM) shown in Figure 8b with only few domains of low skill score (less than 0.8). The skill of the model is comparatively low during December-January-February (DJF) shown in Figure 8a with twelve climatic regions scoring less than 0.8.

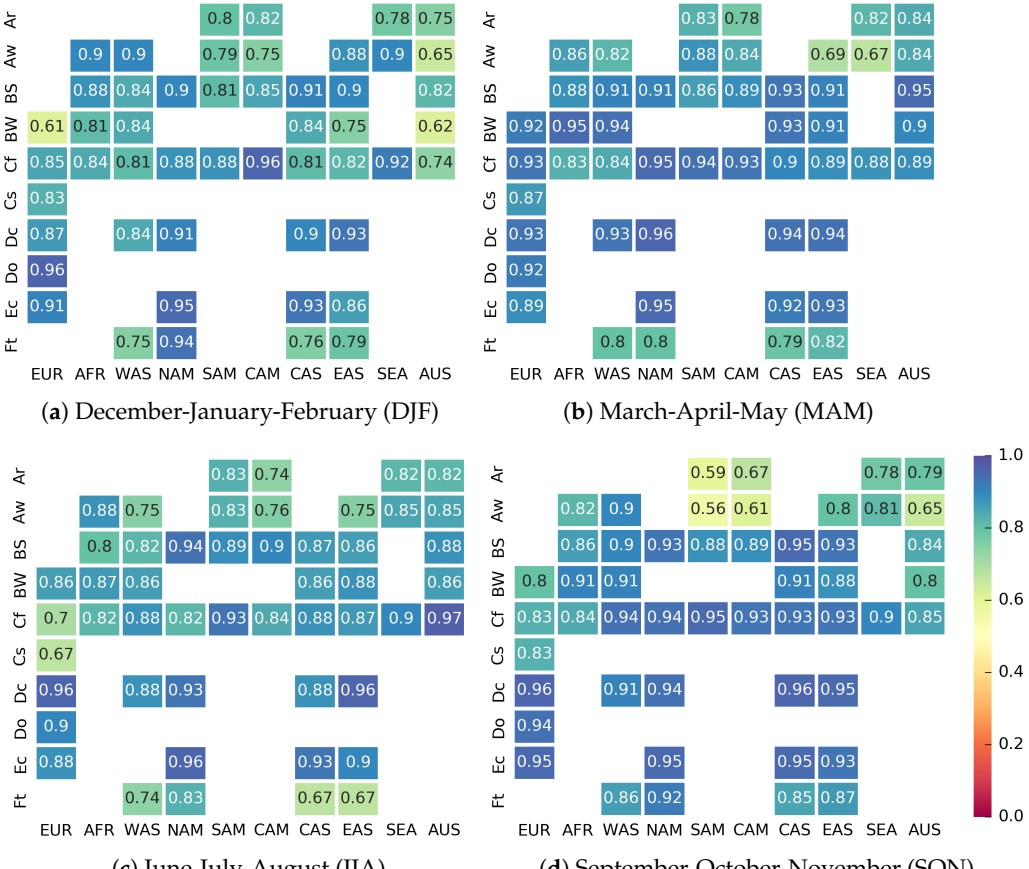

**Figure 8.** Temperature skill scores based on the probability density functions (*Skill$_{seasons}$*) of REMO compared to CRU during the four seasons (DJF, MAM, JJA, SON) for the period 1981 to 2010. The *x*-axis are the domains (Figure 1) while the *y*-axis are the climate types (Table 2). The unit is dimensionless.

The skill of the model in simulating the precipitation decreased compared to the temperature skill scores. More than ten regions have relatively low skills (less than 0.8) with eighteen regions during MAM (Figure 9b). The model has a low skill in tropical humid (Ar) and tundra (Ft) climate regions, especially in terms of precipitation.

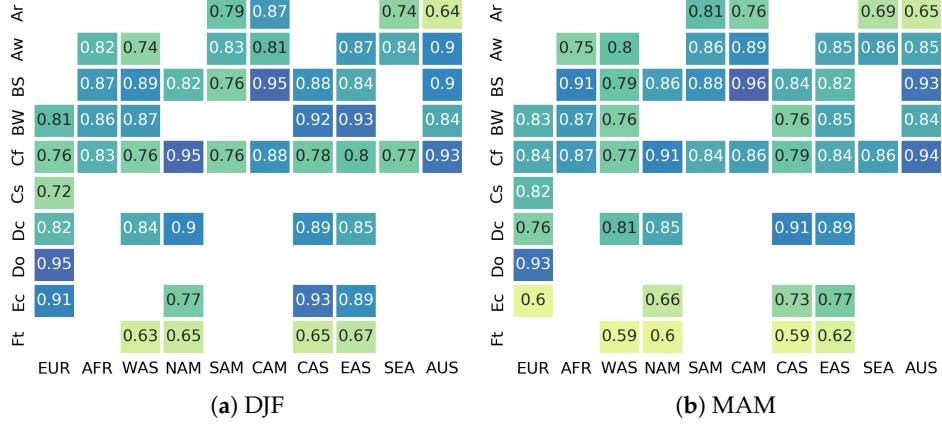

**Figure 9.** *Cont.*

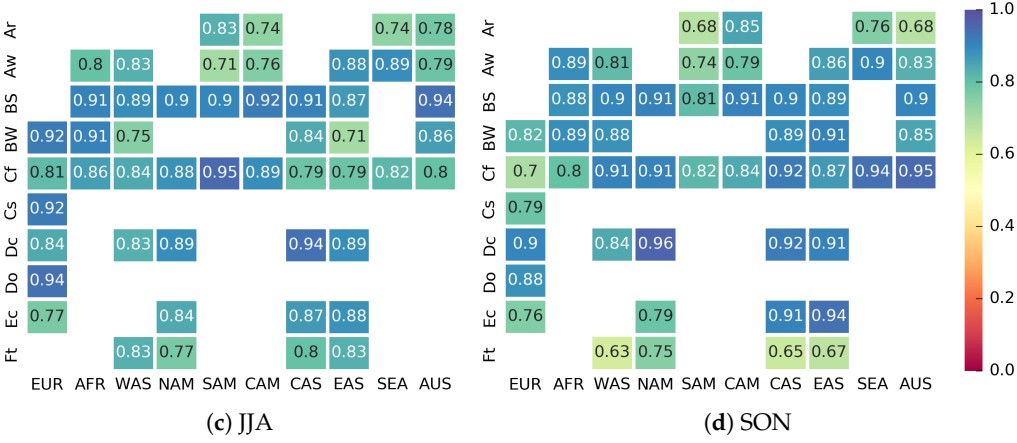

**Figure 9.** The same as Figure 8 but for precipitation.

## 4. Discussion

The model performance of REMO2015 in regions with similar climate types were evaluated in terms of the model biases (Figure 6) and skill score based on probability density functions (Figures 8 and 9). The sources of biases in each climate type regions were investigated and categorized to systematic errors due to missing or misrepresented atmospheric processes or due to observational uncertainty. To explain further the sources of these biases, the mean annual cycles of the current REMO2015 simulations were compared with CRU and other observational datasets available (GLDASD, GPCC, UDEL, and ERAINT). In addition, the previous CORDEX simulations from the old domains (EUR-44, EUR-11, AFR-44, WAS-44, NAM-44, and SAM-44) were compared to the new CORDEX-CORE domains (EUR-22, AFR-22, WAS-22, NAM-22, and SAM-22).

Another source of biases could be inherited by the model from the input boundary forcing on temperature. The CORDEX-CORE simulations in this study were driven by the ERA-Interim reanalysis dataset. To compare the reanalysis with the observational dataset, Figure 10 estimates the mean annual temperature bias of the driving boundary conditions (ERA-Interim) compared to the CRU observational dataset.

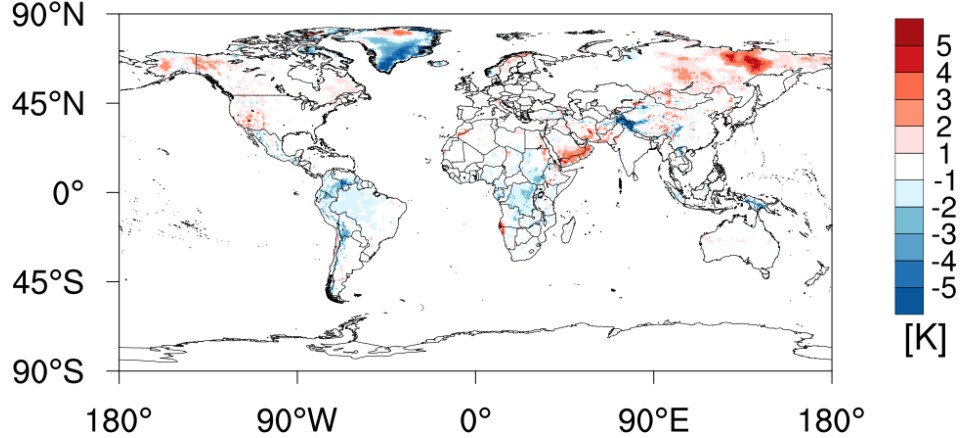

**Figure 10.** Mean annual temperature bias [K] of the ERA-Interim reanalysis (ERAINT) against the Climatic Research Unit (CRU) observational dataset for the period 1981 to 2010.

The biases of ERA-Interim compared to CRU ranges from −5 to +5 K. The warm biases are located in Namib Desert in Africa, coasts of Yemen and Oman, and the eastern regions of Russia. In these regions of Russia, the station density is low [64]; hence, the biases are due to observational uncertainty. The cold biases of ERA-Interim compared to CRU are located in central regions of the African continent, the Himalayas, Indonesia, Bolivia, northern regions of South America, and Greenland. In some cases, the stations used in CRU were insufficient reproducing a known bias such as in the Namib Desert,

near the coast of Africa [65]. The location of these biases are useful, however, in identifying the regions where the REMO simulations have inherent biases from their input driving fields.

In the following subsections, we discuss the sources of errors using the climate types derived from the Köppen–Trewartha Climate Classification.

### 4.1. Tropical Climates

The tropical climate types are found in the seven model domains. In a tropical humid climate (Ar), REMO has a tendency to have a cold and dry bias in SEA while it has a cold and wet bias in AUS. The cold bias in SEA is lower than in AUS. The cold bias especially over the western part of Indonesia is mainly due to the driving input boundary conditions (Figure 10) and as evidently shown in the annual mean cycle of the reanalysis (ERAINT) in Figure 11. In these two domains (AUS and SEA), a cold bias is inherent from the input boundary forcing (ERAINT) and the model (SEA-22/AUS-22 REMO2015) simulated the mean annual cycle closer to the other observational datasets (GLDASD, UDEL), which was still colder by about 1 °C than the CRU dataset. Note that the Ar region in SEA covered a larger area than AUS (see Table 5).

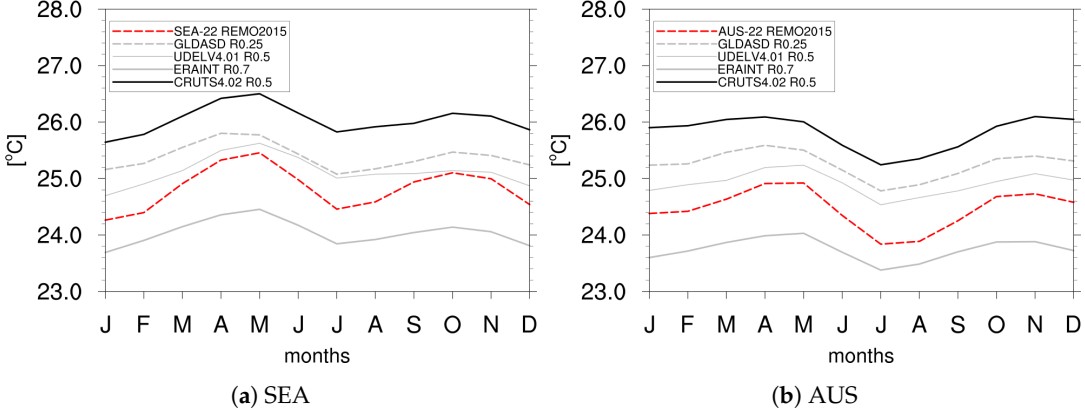

**Figure 11.** Mean annual temperature cycles for tropical humid (Ar) climate type in SEA and AUS for the period from 1981 to 2010. The CRU and other observational (GLDASD, UDEL) and reanalysis (ERAINT) datasets are depicted with black and gray lines. The REMO model (SEA-22/AUS-22 REMO2015) are shown with red lines. The dashed lines indicate similar native resolution (0.22 and 0.25).

The opposing wet and dry bias in Figure 12 are possibly due to the latitude-dependent parameters in these two domains. The AUS domain, which is predominantly located in the mid-latitudes, uses the default parameters for ZDLAND, while the SEA domain uses the modified (larger) ZDLAND for the tropics (Table 1). This effect can also be seen in comparison of the results for SAM (wet bias, standard ZDLAND) and CAM (dry bias, larger ZDLAND). While there are many processes included in the convection scheme, the performance seems to be quite sensitive to ZDLAND. Using a large value for ZDLAND, the rainfall is suppressed because it takes longer to reach a cloud height from which rain can fall. Using the small standard ZDLAND has the opposite effect. A comprehensive study on this feature is needed, which also should include the analysis at daily and sub-daily frequency.

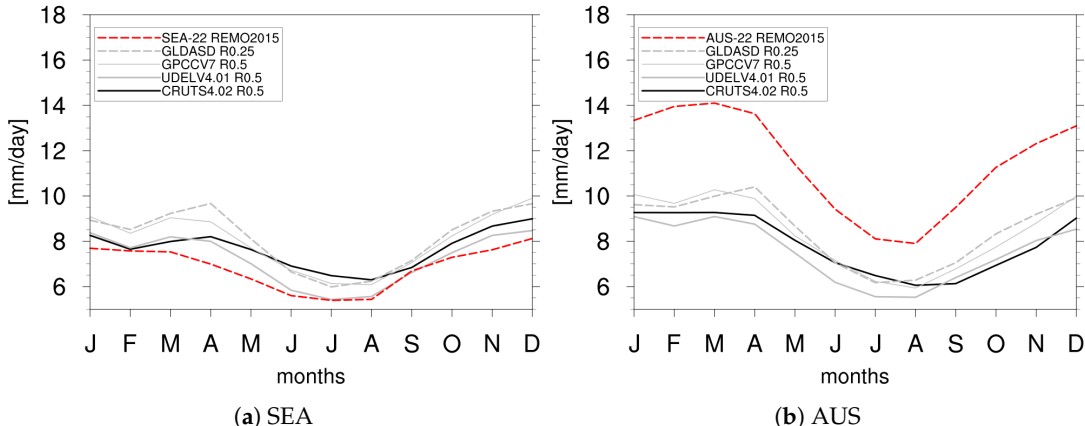

**Figure 12.** Same as in Figure 11 but for the mean annual precipitation cycles. The Global Precipitation Climatology Centre (GPCC) observational dataset is additionally included.

In a tropical with dry "winters" climate (Aw, e.g., CAM in Figure 5a and SAM), the input boundary forcing has a cold mean annual bias in the northern region of South America (Figure 10); however, REMO tends to simulate a warm mean annual bias in both SAM and CAM domains (Figure 6a). Compared to the previous CORDEX study (SAM-44), the CORDEX-CORE simulation (SAM-22) follows the same mean annual cycle of temperature and both SAM-22 and SAM-44 simulations have a warm bias of almost 2 K during austral spring from September to November (figure not shown). Although the land surface parameters were adjusted [54] to represent the Amazon conditions, REMO missed representing the region. This misrepresentation of the land surface conditions could be solved in future studies by using a dynamic vegetation scheme [38]. In addition, REMO has a wet (Figure 13a) bias in SAM only during January to March while it has a slightly dry (Figure 13b) bias in CAM throughout the year. The opposing wet and dry biases in SAM and CAM (Figure 13), respectively, are possibly due to the different latitude-dependent parameters of the two domains (Table 1). In the SAM domain, the wet bias occurs, especially during the austral summer (DJF), where mesoscale convective systems are more active [66].

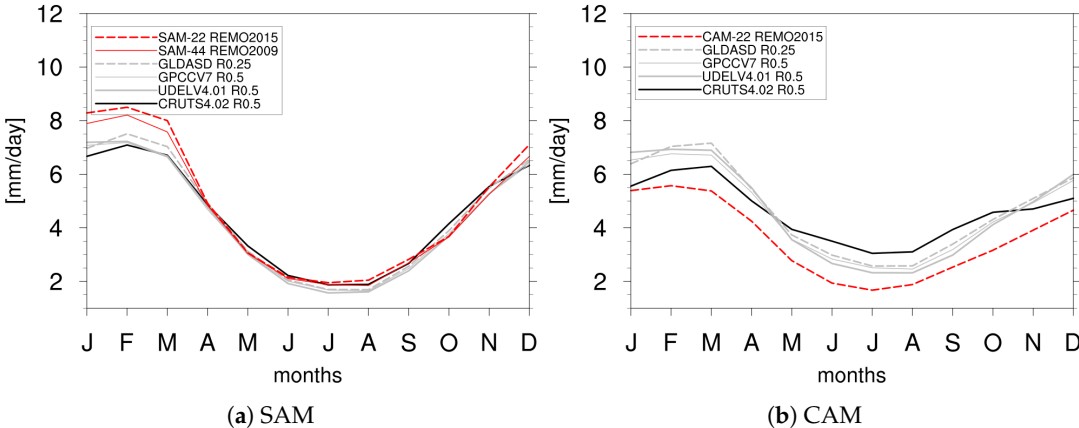

**Figure 13.** Mean annual precipitation cycles for Aw in SAM and CAM. The CRU and other observational (GLDASD, GPCC, UDEL) datasets are depicted with black and gray lines. The REMO model (SAM-44/SAM-22/CAM-22 REMO2015) is shown with red lines. The dashed lines indicate similar native resolution (0.22 and 0.25). Monthly values are averaged over the common period from 1989 to 2008 (1981 to 2010) for SAM (CAM).

## 4.2. Dry Climates

Except for SEA, most of the domains contain the dry climate types. In the regions with a semi-arid (BS) climate type, REMO has generally a warm bias, which indicates some missing processes

misrepresented in this climate type that is a transition between the arid and tropical climate types. REMO has a warm and slightly dry bias in AFR and WAS domains as shown in Figure 6. REMO has a warm and wet bias in NAM, SAM, CAS, EAS, and AUS domains.

In the regions with the arid (BW) climate type or desert regions, REMO has the warmest biases in EAS and AUS as shown in Figure 6a. This warm bias in this climate type region occurred throughout the year (Figure 14). One possible source of this bias is the warm bias of the driving field (ERA-Interim) compared to CRU especially during summer (JJA in EAS and DJF in AUS). A study investigating the CORDEX-Australasia RCM ensemble at 50 km using the Australian Gridded Climate Data [67] showed that most models in the AUS domain show a cold bias in the daily maximum temperature (ensemble mean of about $-1$ to $-2$ K), while the daily minimum temperature was mainly overestimated with some exceptions [63]. A mean temperature comparison was not made due to the limitations of the observational dataset, but the REMO results over Australia were comparable within the range of other RCMs.

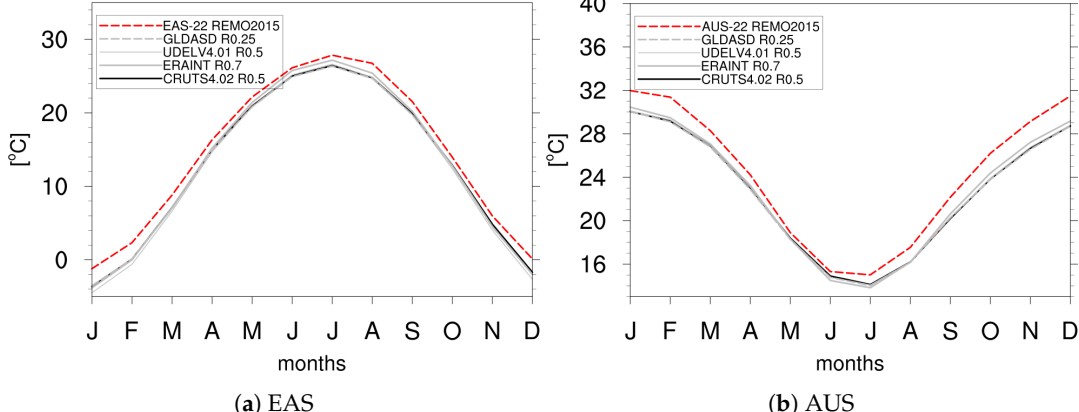

(**a**) EAS　　　　　　　　　　　　　　　　　　　(**b**) AUS

**Figure 14.** Mean annual temperature cycles for arid (BW) climate type in EAS and AUS. The CRU and other observational (GLDASD, UDEL) and reanalysis (ERAINT) datasets are depicted with black and gray lines. The REMO model (EAS-22/AUS-22 REMO2015) is shown with red lines. The dashed lines indicate similar native resolution (0.22 and 0.25). Monthly values are averaged over the period from 1981 to 2010.

The warm and dry bias in EAS corresponds to a missing process over desert regions where the precipitation especially during JJA were underestimated in REMO (Figure 15a). Precipitation is also suppressed with the higher cloud height in EAS compared to AUS, where a wet bias occurs especially during austral summer (DJF), as shown in Figure 15b.

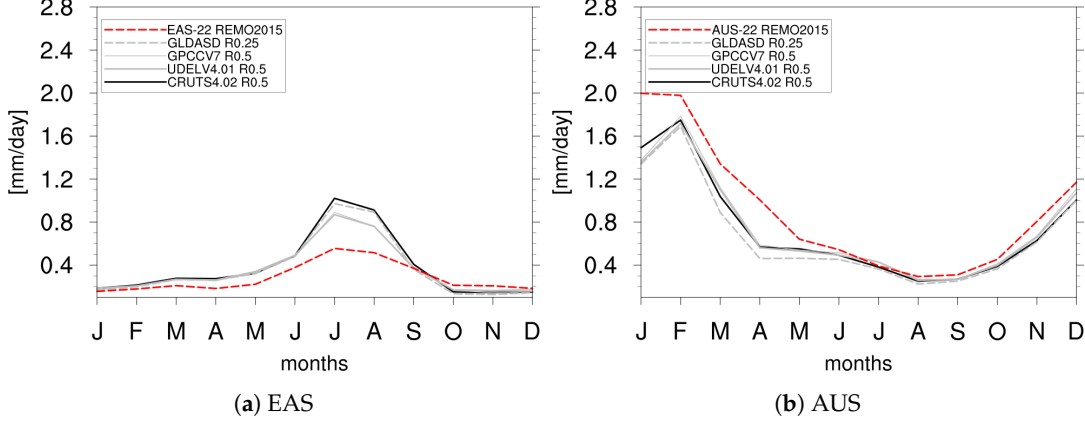

(**a**) EAS　　　　　　　　　　　　　　　　　　　(**b**) AUS

**Figure 15.** Same as in Figure 14 but for the mean annual precipitation cycles. The Global Precipitation Climatology Centre (GPCC) observational dataset is additionally included.

The choice of domain is one of the possible sources of biases. REMO results indicated a cold and wet bias in the regions with BW climate type over EUR domain, but this is mainly located on the African continent, which is located at the southern border of the EUR domain. This cold bias is reduced in the AFR domain; however, the BW region is larger in AFR than in EUR (see Table 5).

For dry regions such as the Sahara desert, the REMO simulations tend to have a cold and wet bias (Figure 3). One of the factors that contribute to this cold bias is too many clouds reproduced by REMO resulting in less net surface solar radiation (figure not shown). Another possible factor is the missing coupled aerosol-cloud interaction, which some other modelling studies recommended to use due to the active dust aerosols in this region (e.g., [68]). This missing interaction could also explain the biases occurring in other deserts such as Namib and Patagonia.

### 4.3. Subtropical Climates

All of the ten model domains contain the subtropical humid (Cf) climate type. REMO-simulated results have a warm and dry bias in EUR, AFR, and WAS domains, but the REMO results have a warm and wet bias in NAM, CAS, AUS, and EAS domains. In SAM and CAM, temperature biases are quite low while the precipitation is overestimated. The precipitation biases are relatively low in this climate zone ranging from −15% to 56%. The highest temperature biases are similar regions in an overlapping domains of WAS (+1.04 K) and EAS (+0.91 K), which was still below the temporal variability (STD) of the domains (see Table 3).

One of the sources of biases in this climate region is possibly due to the large-scale dynamics such as monsoonal processes (e.g. [8]). In WAS, although the skill of the model in simulating the temperature distributions in Cf regions over WAS were relatively high, but it was relatively low in terms of precipitation distributions especially during DJF and MAM. The shift in the precipitation mean annual cycle (Figure 16b) depicts an early onset of the monsoon in the REMO simulations. This early onset of the monsoon could possibly be due to the warm biases of both the REMO simulations (WAS-44 REMO2009 and WAS-22 REMO2015) shown in Figure 16a.

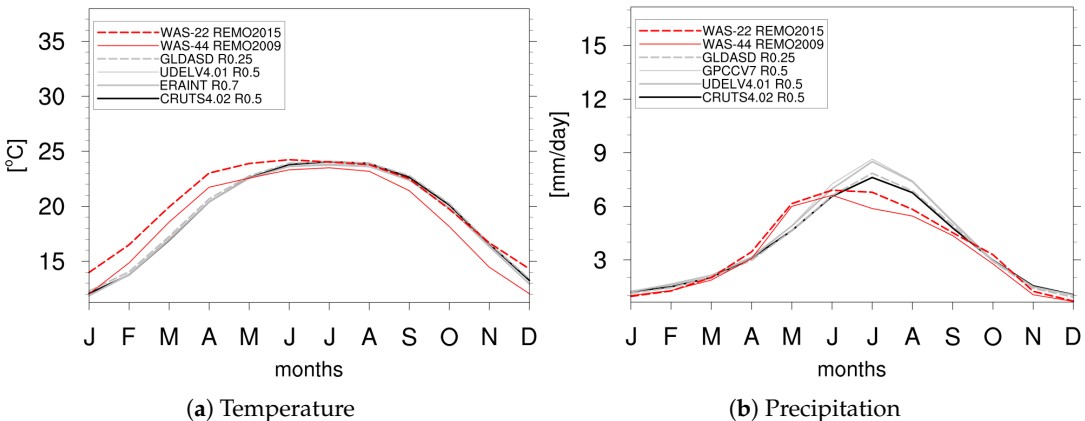

(**a**) Temperature　　　　　　　　　　　　　　(**b**) Precipitation

**Figure 16.** Mean annual temperature and precipitation cycles for subtropical humid (Cf) climate type in WAS for the period from 1989 to 2008. The CRU and other observational (GLDASD, GPCC, UDEL) and reanalysis (ERAINT) datasets are depicted with black and gray lines. The REMO model (WAS-44/WAS-22 REMO2015) are shown with red lines. The dashed lines indicate similar native resolution (0.22° and 0.25°).

In subtropical regions producing a wet summer and dry winter or Cs climate type, this regions are significantly present (area more than 5%) only in the EUR domain. The cold and dry bias of the model compared to the observational datasets is relatively low, and the *Skill_seasons* is relatively high except for summer temperature (Figure 8c), indicating that the model represents the Cs climate type relatively well.

### 4.4. Temperate Climates

The climate of the higher latitude regions is mainly dependent on temperature (thermal zones). The dominant climate type in EUR and NAM is the temperate continental (Dc) climate type, which accounts for more than 25% of its domain land area (Table 5). In EUR, where the model has been intensively developed and already higher resolution of 12.5 km is existing (EUR-11), the mean annual temperature and precipitation biases are relatively low and the PDF skill is relatively high except for precipitation. In regions with the Dc climate type (Figure 17), the summer precipitation averages about 2 mm/day, and the model reproduces excess precipitation throughout the year.

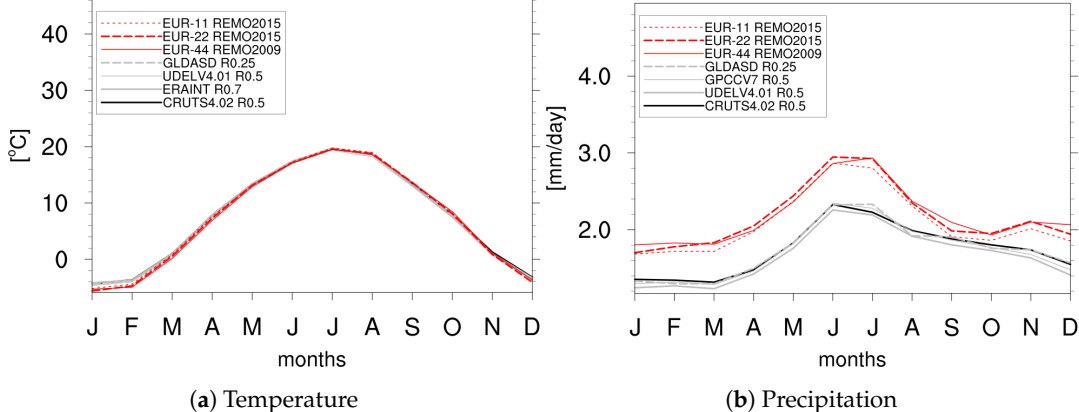

(**a**) Temperature         (**b**) Precipitation

**Figure 17.** Mean annual temperature and precipitation cycles for temperate continental (Dc) in EUR for the common period from 1989 to 2008. The CRU and other observational (GLDASD, GPCC, UDEL) and reanalysis (ERAINT) datasets are depicted with black and gray lines. The REMO model (EUR-44 REMO2009 and EUR-11 REMO2015) is shown with red lines. The dashed lines indicate similar native resolution (0.22° and 0.25°). The dotted line indicate the highest spatial resolution of 0.11°.

For regions located near the ocean, the climate exhibits an oceanic temperate type (Do) and is significantly present (area more than 5%) only in the EUR domain. The warm and wet bias is relatively low and the $Skill_{seasons}$ is relatively high (close to unity) throughout the year (Figures 8 and 9), indicating that the model represents the Do climate type well.

### 4.5. Sub-Arctic or Boreal Climates

The continental sub-arctic or boreal climate type (Ec) occurred in EUR, NAM, CAS, and EAS domains. In these regions, the sources of bias stem from the input boundary forcings (Figure 10). In the EAS domain, Ec is the dominant climate type (more than 19% of the land area) and REMO has a relatively low warm bias of about +0.7 K. The temperature skill of the model measured by its normalized PDF was relatively high although a warm bias of the input driving fields is evident during winter (Figure 18a); however, the precipitation skill is relatively low during winter and spring where the observational uncertainty is rather high (especially during JJA) as shown in Figure 18b. In these climate regions, the precipitation amount is relatively low and station density in the low-populated mountainous and desert areas of the west and northwest of China [69].

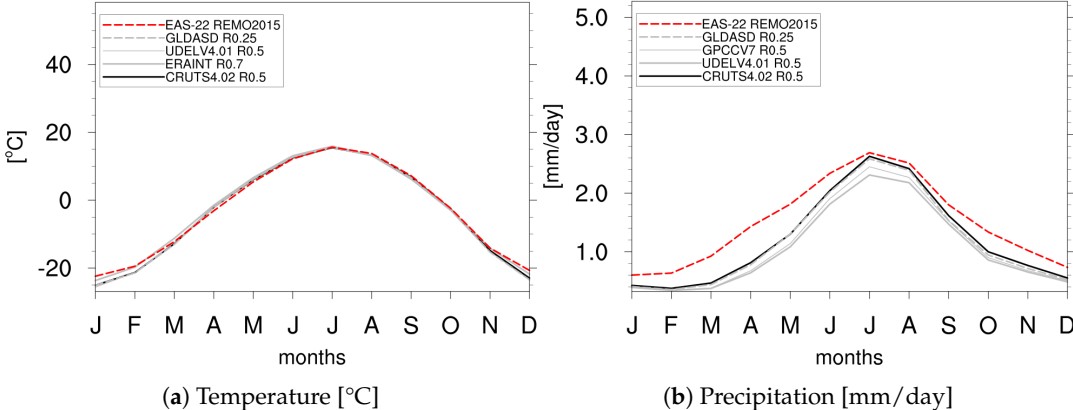

**(a)** Temperature [°C]      **(b)** Precipitation [mm/day]

**Figure 18.** Mean annual temperature and precipitation cycles for boreal (Ec) climate type in EAS for the period from 1981 to 2010. The CRU and other observational (GLDASD, GPCC, UDEL) and reanalysis (ERAINT) datasets are depicted with black and gray lines. The REMO model (EAS-22 REMO2015) is shown with red lines. The dashed lines indicate similar native resolution (0.22° and 0.25°).

*4.6. Polar Climates*

The polar climates with tundra or highland (Ft) are located in WAS, NAM, CAS, and EAS domains. The high mountain regions of Himalayas are present in the WAS, CAS, and EAS domains, where REMO has a very cold and very wet bias. One main source of the cold bias is due to the ERA-Interim boundary forcing, which already has a large cold bias of more than −4 K (Figure 10). Although the biases were within the range of the domain temporal and spatial variability (Tables 3 and 4), the low skill of simulating the temperature and precipitation PDFs is largely attributed to the complex orography. In these regions, the model tends to produce the cold bias of and wet bias throughout the year (Figure 19). Another known source of error is the observed precipitation undercatch in high latitudes and mountainous regions where errors could reach up to 80% [70]. Although some global datasets such as GPCC attempted to reduce this error, undercatch correction remains a challenge in global datasets due to unavailable important information such as gauge characteristics and exposure [71]. Further analysis of these regions' station data or other available high resolution observational datasets would be needed in these regions.

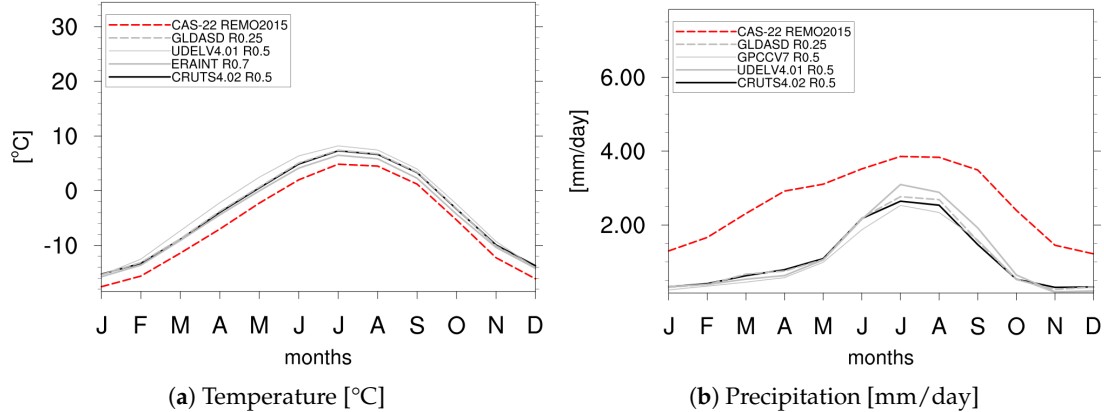

**(a)** Temperature [°C]      **(b)** Precipitation [mm/day]

**Figure 19.** Same as Figure 18 but for tundra (Ft) climate type in Central Asia (CAS).

**5. Conclusions**

In this study, the ability of the latest REMO regional climate model (REMO2015) to simulate the present climate in all inhabited regions world-wide was quantified in terms of biases and skill compared with observations within the CORDEX-CORE framework. The high resolution simulations of 0.22° (about 25 km) were driven by the ERA-Interim reanalysis over the following domains: Europe, Africa, South Asia, North America, South America, Central America, Central Asia, East Asia, Southeast

Asia, and Australasia and the period of analysis is from 1981 to 2010. The new datasets were also compared with existing simulations of REMO from the CORDEX framework. The selected regions of analysis were based on the Koeppen–Trewartha Climate Classfication where ten of the fourteen climate types were used.

Based on the biases (Figure 6) and seasonal skill (Figures 8 and 9), REMO2015 represents precipitation and temperature relatively well in most climate type regions of EUR, NAM, CAM, and SEA. The simulated temperature and precipitation in subtropical (Cf, Cs) and temperate (Dc, Do) climate regions have relatively low biases (about ±0.5 K and ±50%). The low biases especially in EUR indicate that the model has been intensively used and developed in this domain; however, most of the model biases in the other domains are within the range of the observed variability (see Tables 3 and 4).

The sources of model biases and low model skills were identified as missing or misrepresented processes in tropical and dry regions, observational uncertainty especially in less populated regions such as in polar regions and high mountains, and inherent biases from the input boundary forcing. The discussion on the input boundary forcing was crucial in identifying the regions where the ERA-Interim already has an inherent bias compared to the CRU. Two of the challenging domains were the CAS and EAS, where the cloud height parameter (ZDLAND) was increased compared to the default values. In these two large domains, the simulated temperature and precipitation in regions with dry climates (BW) were too warm and slightly dry, which indicated a suppressed production of rainfall and warmer atmosphere. As discussed in regions with tropical humid (Ar) climate types (Section 4.1), more cloud water is needed for the formation of rainfall using the large values for ZDLAND in SEA compared to small values for ZDLAND in AUS. Further sensitivity studies are recommended in these large domains that contain regions with several climate types, especially in the overlapping regions. In regions with polar climates (Ft), the simulated temperature and precipitation were very cold and very wet, but this was possibly due to the input boundary forcing and observational uncertainty discussed in Section 4.6.

In order to decouple the sources of biases for each domain, further sensitivity studies are planned to tackle the regions with high model biases especially in the tropical and polar climates. A regional analysis for each domain is needed in pinpointing the possible causes of biases and low model skill such as misrepresented process of the flow regimes, environmental effects (e.g., topography), and possible missing land-atmosphere-ocean processes (e.g., vegetation feedbacks, coastal regions). Investigating the biases of the model in each domain could facilitate further development of the model. However, as shown in this present study, quantifying the REMO model biases and skill provided the necessary information to identify the regions where the model has a high or low skill. We have seen that REMO is able to sufficiently represent the climate of the regions using the Köppen–Trewartha climate classification and could be further used for climate change studies.

**Author Contributions:** The individual contributions are as follows: Conceptualization: A.R.R., C.T., K.S., and D.J.; methodology: A.R.R., L.B., and K.S.; software: A.R.R., L.B., K.S., and T.W.; validation, A.R.R., C.T., and T.W.; formal analysis: A.R.R., C.T., L.B., K.S., T.W., D.R., P.H., C.N., L.K., and D.J.; investigation: A.R.R., C.T., L.B., K.S., T.W., D.R., P.H., C.N., L.K., and D.J.; resources: D.R., K.S., and D.J.; data curation: L.B., A.R.R., and K.S.; writing—original draft preparation: A.R.R.; writing—review and editing: A.R.R., C.T., L.B., K.S., T.W., D.R., P.H., C.N., L.K., and D.J.; visualization: A.R.R., T.W., and K.S.; supervision, D.J.; project administration, K.S. and C.T.; funding acquisition, D.J.

**Funding:** This research received no external funding.

**Acknowledgments:** The CORDEX-CORE REMO simulations were performed under the GERICS/HZG share at the German Climate Computing Centre (DKRZ). The authors are also thankful for the DKRZ support in a variety of areas such as data storage, post processing, cmorizing using cdo-cmor, and the use of analysis tools such as jupyternotebook. Some work was conducted within the framework of the Helmholtz Institute for Climate Service Science (HICSS). The work related to the Central Asian domain (CAS) was conducted in the frame of the AFTER project supported by the Federal Ministry of Education and Research (BMBF); AFTER is granted by the ERA.Net RUS Plus Initiative, ID 166. UDel AirT Precip data was provided by the NOAA/OAR/ESRL PSD, Boulder, CO, USA, from their Web site at https://www.esrl.noaa.gov/psd/. We are grateful for the internal review from Katharina Buelow and for the insights and recommendations from four anonymous reviewers to improve our manuscript.

**Conflicts of Interest:** The authors declare no conflict of interest.

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
