# Peer review of "Evaluation of New CORDEX Simulations Using an Updated Köppen–Trewartha Climate Classification"

_atmosphere, doi:10.3390/atmos10110726_

Round 1
Reviewer 1 Report
The paper presents the results of the REMO2015 climatic model for 10 different global regions at 25 km in the present climate which are evaluated against CRU gridded observations employing temperature and precipitation. Apart from the “traditional” manner of deriving biases of both parameters, the authors made an attempt to analyse the biases for different climatic types as derived from Köppen-Trewartha Climate Classification and subsequently to discuss on possible sources of biases. For this reason, I found the paper gets a better insight in the climatic modeling studies, providing more than simple statistical metrics using a well- known climatic model with a better resolution.
I have the following comments:
Abstract: The statements in lines 1-4 are not necessary for the abstract. They could only state instead “Within the context of CORDEX-CORE experiments ……”. Why the projections of REMO model are stated? The objective of the study is not clarified. Lines 48-54: this paragraph is very confusing. The authors mention runs within the context of CORDEX-CORE and then (if I understand well) they refer to different (??) results at a resolution of 25 km for different (?) regions, which are 10 out of 14. Why are 10 and not 14?What do they mean by “few ensemble members”? What are the “new simulations”? Are there new simulations within the scope of the present paper? The abbreviations that are used in the study for the 10 regions should be displayed in Figure 1 (or in a similar plot). This will facilitate the reader to follow the results and the discussion. Lines 63-70: the objective should be more clearly stated. Lines 90-98: I think that the use of references in this paragraph is not consistent with the guidelines of the journal Line 104: why the specific unit conversion is numbered as an equation? Lines 128-131:Following previous comment, I got confused with the different regions and the abbreviations. Finally, what are the regions used in the present study? The reader does not mind about the regions used in the CORDEX –CORE or in other projects. Lines 141-143: What is the meaning of this statement? Please clarify. Lines 165-166: I got confused. Are the runs used in the present study non hydrostatic? In section 2 the authors mentioned that REMO is hydrostatic model. Section 3.1: the authors do not comment more thoroughly on the poor performance of the model in some regions, such as eastern Africa for precipitation. Why the absolute bias is used for temperature and relative bias for precipitation? Similarly in Tables 3 and 4. Are there significant improvement of the results of REMO 2015 as compared to previous versions of REMO or with the same model and lower resolution? Lines 264-265: why do the authors mean “Note that the threshold used in determining the climate types …….. In future studies, …., such as Aw, BS, and BW (Figure 5)”. What atmospheric flows do they mean? Why specially in Aw, BS, BW? Figure 6: why these tables are called “heat maps”? Section 3: how the bias in Figure 6 is calculated? Is the “bias” derived from the comparison between the two climatologies? Section 4: I got the impression that the only sources of bias are driving input boundary conditions and observational uncertainty or at lesser extent some model parameters such as ZDLAND. However, there many other potential sources related to the model such as topography (that is not adequately represented in a climatic model), microphysics or other dynamic processes that are not discussed in depth. Section 5: Are there any other studies where the climatic types are used as a basis for model evaluation? How innovative are these results for climatic modeling and future projections?Author Response
Dear Reviewer,
Thank you very much for your insights and comments to improve our manuscript. We took the liberty in numbering your comments to facilitate our discussion.
1. Abstract: The statements in lines 1-4 are not necessary for the abstract. They could only state instead “Within the context of CORDEX-CORE experiments ……”. Why the projections of REMO model are stated? The objective of the study is not clarified.
The premise of the WCRP CORDEX-CORE Framework is essential to introduce the upcoming high resolution simulations based on two regional climate models. The simulations include hindcast driven by a reanalysis and future climate projections driven by 3 GCMs and on two RCPs. However, in this study, we only focus on the evaluation of the hindcast simulation of the REMO2015 model. We rephrased the opening statements: “A new ensemble of climate and climate change simulations covering all major inhabited regions with a spatial resolution of about 25 km, from the WCRP CORDEX COmmon Regional Experiment (CORE) Framework, has been established in support of the growing demands for climate services. The main objective of this study is to assess the quality of the simulated climate and its fitness for climate change projections by REMO (REMO2015), a regional climate model (RCM) of Climate Service Center Germany (GERICS) and one of the RCMs used in the CORDEX-CORE Framework. The CORDEX-CORE REMO2015 simulations were driven by the ECMWF ERA-Interim reanalysis and the simulations were evaluated in terms of biases and skill scores over ten CORDEX Domains against the Climatic Research Unit (CRU) TS version 4.02, from 1981 to 2010 according to the regions defined by the Köppen-Trewartha (K-T) Climate Classification types.”2. Lines 48-54: this paragraph is very confusing. The authors mention runs within the context of CORDEX-CORE and then (if I understand well) they refer to different (??) results at a resolution of 25 km for different (?) regions, which are 10 out of 14. Why are 10 and not 14?What do they mean by “few ensemble members”? What are the “new simulations”? Are there new simulations within the scope of the present paper? The abbreviations that are used in the study for the 10 regions should be displayed in Figure 1 (or in a similar plot). This will facilitate the reader to follow the results and the discussion.
Within the previous CORDEX Framework, 14 domains were constructed. Except for Europe (0.11 degree or 12.5 km resolution) and Southeast Asia (0.22° or 25 km resolution), all the other 12 domains were run at 0.44° or 50 km resolution. In the new CORDEX-CORE Framework, only 10 domains out of the existing 14 domains were used and the simulations were designed at a spatial resolution of 0.22°. The other four domains (Antarctica, Arctic, Meditteranean, and the MENA Region) were excluded due to low populated regions or redundancy (e.g. Meditteranean region is already in Europe domain). The few ensemble members are referred to previous studies e.g. in Central Asia (Ozturk et al., 2012) and Central America (Fuentes-Franco et al., 2015), where the regional climate change assessments are either single model or few ensemble members (at least 4). We added references to these studies. The new simulations referred in this study are those from the new CORDEX-CORE Framework, which include hindcast (evaluation) and climate change (rcp2.6 and rcp8.5) simulations. Only the hindcast simulations are part of this paper to evaluate the REMO model. The lines 48-54 were simplified. “Within CORDEX-CORE [12], new high resolution simulations of 0.22° (about 25 km) were setup over most domains, except for Southeast Asia [13,14] and Europe [15] which were already at this resolution or higher (at about 12.5 km over Europe). These new high resolution simulations will provide additional climate simulations over regions especially with few ensemble members e.g. over Central Asia [16] and Central America [17]. Figure 1 shows ten out of the fourteen CORDEX domains that were used in this study.” We also modified Figure 1 to include domain labels.3. Lines 63-70: the objective should be more clearly stated.
4. Lines 90-98: I think that the use of references in this paragraph is not consistent with the guidelines of the journal.
We updated the reference lists: Tiedtke, M. A Comprehensive Mass Flux Scheme for Cumulus Parameterization in Large-Scale Models. Monthly Weather Review 1989, 117, 1779–1800. doi:10.1175/1520-0493(1989)117<1779:ACMFSF>2.0.CO;2. Nordeng, T.E. Extended versions of the convective parametrization scheme at ECMWF and their impact on the mean and transient activity of the model in the tropics. Technical report, ECMWF Research Department, European Centre for Medium Range Weather Forecasts, Reading, UK, 1994.5. Line 104: why the specific unit conversion is numbered as an equation?
We rephrased the equation within the paragraph: “The REMO standard value for ZDLAND, which was originally developed over Europe, is 750 m · g where g is the acceleration due to gravity and assuming a cloud height of 750 m. The tropical value usually reaches the height of 1500 m or 3000 m.”6. Lines 128-131:Following previous comment, I got confused with the different regions and the abbreviations. Finally, what are the regions used in the present study? The reader does not mind about the regions used in the CORDEX –CORE or in other projects.
These lines were moved to an earlier paragraph (near Table 1) to clarify the domains used in this study as well as the parameters used. “In this study, the REMO model was used to simulate the ten domains shown in Figure 1. The other model parameters such as number of grid boxes (x and y), minimum cloud height over land (ZDLAND-L) or ocean (ZDLAND-O) for which rain can fall, and the Charnock constant (CHAR) for each domain are listed in Table 1. In this setup, the largest domains were Africa (AFR), East Asia (EAS), and Australasia (AUS) while the smallest domains were Europe (EUR) and Southeast Asia (SEA). Since ZDLAND is latitude-dependent, different values were set for domains covering the tropics and the domains covering the mid-latitudes. CHAR is also a parameter that can vary in different regions due to its dependency of the general characteristics of the sea conditions.”7. Lines 141-143: What is the meaning of this statement? Please clarify.
In characterizing the regions to climate types, we are ensuring that each grid box has only one climate type. We start with the polar and boreal climates (Types F and E) followed by the dry climate (Type B) and the temperate climate (Type D). These climate types are mainly driven by temperature. For the rest of the climate types, which are mainly driven by both temperature and precipitation, we select the subtropical climate (Type C) and then the tropical climate (Type A). We revised the lines to: “In order to make sure that each grid box have exclusive climate types, the order of obtaining climate types are the following: the polar and boreal climates (Types F and E), followed by the dry climate (Type B), the temperate climate (Type D), and then, subtropical and tropical climates (Types C and A).”
8. Lines 165-166: I got confused. Are the runs used in the present study non hydrostatic? In section 2 the authors mentioned that REMO is hydrostatic model.
9. Section 3.1: the authors do not comment more thoroughly on the poor performance of the model in some regions, such as eastern Africa for precipitation.
We presented the overall results of temperature and precipitation biases on Section 3.1. The sources of biases were discussed thoroughly in Section 4. We attempted to present the results over the ten domains and their sources of uncertainties. We stated the reasons of poor performance of the model which was mainly due to the input boundary forcing, missing or misrepresented processes in tropical and dry regions, and observational uncertainty especially in less populated regions such as in polar regions and high mountains. However, some specific regions like the dry bias for Eastern Africa, we stated that this dry bias relate to the warm bias in Figure 3a. A detailed analysis for specific regions are beyond the scope of this study and will be planned for specific regional analysis as stated in our outlook. “A regional analysis for each domain is needed in pinpointing the possible causes of biases and low model skill such as misrepresented process of the flow regimes, environmental effects (e.g. topography), and possible missing land-atmosphere-ocean processes (e.g. vegetation feedbacks, coastal regions). Investigating the biases of the model in each domain could facilitate further development of the model.”10. Why the absolute bias is used for temperature and relative bias for precipitation?
The absolute bias of temperature reflect the biases of temperature which varies from -5 to 5 K. For precipitation, the relative bias reflect the biases relative to the observed precipitation. For example, an absolute bias of 1 mm/day could mean from a range of 1 % to 100% relative bias compared to the original value. For inhabitants living in wet regions, small absolute biases could reflect small relative biases. However, for inhabitants living in dry regions, small absolute biases could mean 100% change from the observed value. We feel that the relative bias for precipitation provide a meaningful quantitative description in estimating the biases of the regions.11. Similarly in Tables 3 and 4. Are there significant improvement of the results of REMO 2015 as compared to previous versions of REMO or with the same model and lower resolution?
We constructed supplementary tables for Table 3 and 4. The table includes the different domain resolution and model version for the common evaluation period from 1989 to 2008. In this comparison, only the following domains are used: EUR, NAM, AFR, SAM, and WAS. In terms of temperature (Table 3 supplementary), the magnitude of the absolute bias reduces as the resolution increases, except for the NAM domain. In terms of precipitation, the relative bias is almost similar in most domains except for EUR and WAS domains, where the increase in resolution decrease the magnitude of the relative bias. Table 3 Supplementary (1989-2008)
Domains |
CRU mean |
Temporal STD |
REMO bias |
Spatial STD |
EUR-44 (REMO2009) |
11.491 |
7.6977 |
-0.47326 |
1.1416 |
EUR-22 (REMO2015) |
-0.45128 |
1.1319 |
||
EUR-11 (REMO2015) |
-0.33911 |
1.2174 |
||
NAM-44 (REMO2009) |
4.4111 |
9.8122 |
0.10854 |
1.2648 |
NAM-22 (REMO2015) |
0.33901 |
1.259 |
||
AFR-44 (REMO2009) |
9.8122 |
4.2129 |
0.34684 |
1.4702 |
AFR-22 (REMO2015) |
0.25384 |
1.4574 |
||
SAM-44 (REMO2009) |
22.049 |
1.9106 |
0.60079 |
1.246 |
SAM-22 (REMO2015) |
0.59907 |
1.2185 |
||
WAS-44 (REMO2009) |
19.266 |
5.4668 |
-0.99681 |
1.7965 |
WAS-22 (REMO2015) |
0.31732 |
1.9357 |
Domains |
CRU mean |
Temporal STD |
REMO bias |
Spatial STD |
Rel bias |
Rel spatial STD |
EUR-44 (REMO2009) |
1.4973 |
0.86585 |
0.16347 |
0.55617 |
10.195 |
25.151 |
EUR-22 (REMO2015) |
0.084546 |
0.64541 |
5.6392 |
27.665 |
||
EUR-11 (REMO2015) |
-0.033225 |
0.69059 |
0.18902 |
29.085 |
||
NAM-44 (REMO2009) |
1.8475 |
1.1409 |
0.50007 |
0.71979 |
42.141 |
56.559 |
NAM-22 (REMO2015) |
0.49281 |
0.81256 |
40.8 |
58.12 |
||
AFR-44 (REMO2009) |
1.6227 |
1.4599 |
-0.074331 |
0.68562 |
15.776 |
186.0 |
AFR-22 (REMO2015) |
-0.067915 |
0.73285 |
18.534 |
201.96 |
||
SAM-44 (REMO2009) |
4.2889 |
2.7746 |
0.73507 |
2.2783 |
39.138 |
105.07 |
SAM-22 (REMO2015) |
0.84199 |
2.4572 |
40.905 |
111.89 |
||
WAS-44 (REMO2009) |
1.9116 |
1.727 |
-0.014322 |
1.3762 |
14.976 |
128.14 |
WAS-22 (REMO2015) |
0.043981 |
1.446 |
6.73 |
98.633 |
12. Lines 264-265: why do the authors mean “Note that the threshold used in determining the climate types …….. In future studies, …., such as Aw, BS, and BW (Figure 5)”. What atmospheric flows do they mean? Why specially in Aw, BS, BW?
For this paper, as an overview of the biases and skill of the model in simulating ten CORDEX-CORE domains, aggregating the regions into climate types is sufficient especially on dominant climate types such as arid (BW) or continental boreal (Ec) climate. In these climate types, the regions are more clustered. However, in some climate types e.g. BS (semi-arid dry climate), the regions are scattered since they transition from dry climates to subtropical climates. Within this transitional climate types, further analysis is recommended to attribute the sources of biases due to physical processes such as the atmospheric flows. We modified the following lines to incorporate the comments: “A few of the climate masks are shown in Figure 5. Note that the threshold used in determining the climate types are only dependent on the climatological mean of temperature and precipitation. The regions of analysis are aggregated into similar climate types where the dominant climate types such as arid (BW) or continental boreal (Ec) climate are clustered together. However, in some climate types e.g. BS (semi-arid dry climate), the regions are scattered since they transition from dry climates to subtropical climates. Within this transitional climate types, further analysis is recommended to consider physical processes such as the atmospheric flows. Regions influenced by the same atmospheric flow, as well as overlapping regions (e.g. BW in both WAS and CAS domains in Figures 5c and 5d), should be analyzed further.”13. Figure 6: why these tables are called “heat maps”?
The heatmap is a relatively new terminology (https://en.wikipedia.org/wiki/Heat_map#Examples) but it has been used in papers such as this https://www.nature.com/articles/nmeth.1902.14. Section 3: how the bias in Figure 6 is calculated? Is the “bias” derived from the comparison between the two climatologies?
We calculated the absolute bias based on the simulated monthly values subtracted by the observed monthly values. For the relative bias, we normalized the absolute bias with the observed monthly values. The mean bias is the mean of 30 years (in the case of Figure 6). We revised the lines regarding the calculation of bias in Section 2.3 as follows: “The absolute bias of the simulated monthly precipitation and temperature were calculated compared to the observed monthly values. For the relative bias, the mean absolute bias was normalized with the observed climatological mean. The mean absolute and relative biases were then aggregated into regions with similar climate types defined in Table 2.”15. Section 4: I got the impression that the only sources of bias are driving input boundary conditions and observational uncertainty or at lesser extent some model parameters such as ZDLAND. However, there many other potential sources related to the model such as topography (that is not adequately represented in a climatic model), microphysics or other dynamic processes that are not discussed in depth.
Similar to our reply in your comment number 9, we attempted to present the results over the ten domains and their sources of uncertainties. We stated the reasons of poor performance of the model which was mainly due to the input boundary forcing, missing or misrepresented processes in tropical and dry regions, and observational uncertainty especially in less populated regions such as in polar regions and high mountains. A detailed analysis for specific regions are beyond the scope of this study and will be planned for specific regional analysis as stated in our outlook. “A regional analysis for each domain is needed in pinpointing the possible causes of biases and low model skill such as misrepresented process of the flow regimes, environmental effects (e.g. topography), and possible missing land-atmosphere-ocean processes (e.g. vegetation feedbacks, coastal regions). Investigating the biases of the model in each domain could facilitate further development of the model.”16. Section 5: Are there any other studies where the climatic types are used as a basis for model evaluation? How innovative are these results for climatic modeling and future projections?
We provided several references on the use of the Koeppen-Trewartha climate classification in: Assessing climate change effects in Europe from an ensemble of nine regional climate models [De Castro, et al., 2007] Assessing the transferability of REMO to CORDEX Regions [Jacob, et al., 2013], Analyzing climate projections in South America [Fernandez, et. al, 2017], Evaluating climatic refugia in statistically derived ecoregions for the People's Republic of China [Baker et al, 2009]. In addition, other climate classification schemes such as the Koeppen-Geiger have been used in evaluating models and analyzing future climate projections: Engelbrecht et al. (2016): „Shifts in Köppen-Geiger climate zones over southern Africa in relation to key global temperature goals“ Theor Appl Climatol,123:247–261 DOI 10.1007/s00704-014-1354-1. Ayoub Zeroual et al. (2019): „Assessment of climate change in Algeria from 1951 to 2098 using the Köppen–Geiger climate classification scheme“, Climate Dynamics, 52:227–243 https://doi.org/10.1007/s00382-018-4128-0 The use of these climate types are innovative because instead of using political boundaries, we are using regions defined by physical climate characteristics. As mentioned in our introduction, the Köppen-Trewartha (K-T) Climate Classification “have been widely used in previous studies [21-24] due to its similarities with the native vegetation”.Best regards,
Armelle and co-authors
Reviewer 2 Report
The manuscript has conducted quite sufficient work, evaluating simulations produced by REMO2015 for over 10 CORDEX domain. Based on my understanding, below is some points should be handled before accepting this manuscript.
In “Abstract” part, from line 5-10, I think these two sentences should be combined or reorganized for avoiding confusion.
There are several typographical and grammatical errors that need to be corrected. For example:
In line 25, “…World Climate Research Program (WCRP) Initiative on COordinated Regional Downscaling EXperiments or CORDEX”;
In line 134, “The fourteen climate types (Table 2) were derived using a 30-year climatology from an observational dataset to derived the mean annual temperature (Tann, in °C) and the mean annual precipitation (Pann, in cm)”.
In Figure 1, the legend name and unit should be provided appropriately.
The author should explain or provide more information about why you decided to use climate type instead of political boundaries as regions’ classification.
Why the skill score was chosen to be one of indexes to evaluate the performance of REMO2015?
The quality of Figure 7 needs to be improved such as the integrality of y axis for (b) and (d).
In “Conclusion” part, from line 459 to 461, how do you identify the sources of model biases and low model skills?

Author Response
Dear Reviewer,
Thank you very much for your insights and comments to improve our manuscript. We took the liberty in numbering your comments to facilitate our discussion.
1. In “Abstract” part, from line 5-10, I think these two sentences should be combined or reorganized for avoiding confusion.
We modified the lines to avoid confusion: “A new ensemble of climate and climate change simulations covering all major inhabited regions with a spatial resolution of about 25 km, from the WCRP CORDEX COmmon Regional Experiment (CORE) Framework, has been established in support of the growing demands for climate services. The main objective of this study is to assess the quality of the simulated climate and its fitness for climate change projections by REMO (REMO2015), a regional climate model of Climate Service Center Germany (GERICS) and one of the RCMs used in the CORDEX-CORE Framework.”2. There are several typographical and grammatical errors that need to be corrected. For example:
In line 25, “…World Climate Research Program (WCRP) Initiative on COordinated Regional Downscaling EXperiments or CORDEX”;
CORDEX is an acronym and its definition is not a typographical error.In line 134, “The fourteen climate types (Table 2) were derived using a 30-year climatology from an observational dataset to derived the mean annual temperature (Tann, in °C) and the mean annual precipitation (Pann, in cm)”.
Line 134 is revised as “The fourteen climate types (Table 2) were derived using the mean annual temperature (Tann, in °C) and the mean annual precipitation (Pann, in cm) of a 30-year climatology from an observational dataset.”In Figure 1, the legend name and unit should be provided appropriately.
3. The author should explain or provide more information about why you decided to use climate type instead of political boundaries as regions’ classification.
We modified the following paragraph: “The regions of the world can be subdivided into political boundaries or into different climate types based on its long-term precipitation and temperature characteristics. Defining the regions of analysis according to climate types, rather than political zones, provide regional information based on physical processes. Using threshold values of temperature and precipitation, the global climate was originally classified by Köppen [18]. A modification of these thresholds and using an updated observational datasets produced classification schemes such as the Köppen-Trewartha (K-T) Climate Classification [19,20]. The similarities of the latter to the classical scheme were thoroughly discussed in Belda et al [21]. The K-T climate types have been widely used in previous studies [22–25] due to its similarities with the native vegetation. In this study, the K-T climate types were used to define the regions of analysis using an updated observational dataset.”4. Why the skill score was chosen to be one of indexes to evaluate the performance of REMO2015?
In aggregating the grid points from the different regions with similar climate types, the inherent distribution should be defining the climate of the region. To measure the distribution, the skill score using the PDFs give a robust comparison of the observed and simulated precipitation and temperature distribution. We added the following lines [Lines 72-76]: “The skill of the model is quantified by using probability density functions (PDF) of the observed and simulated temperature and precipitation aggregated at each climate type and each region following the PDF skill score method[31]. This method provides a robust comparison of the similarity between the PDF of the simulated and observed values.”5. The quality of Figure 7 needs to be improved such as the integrality of y axis for (b) and (d).
We improved the quality of Figure 7 by showing only 1 example of a climate type mask and its normalized observed and simulated PDFs.6. In “Conclusion” part, from line 459 to 461, how do you identify the sources of model biases and low model skills?
In the discussion part, we elaborated on the sources of model biases and low model skills. First, the inherent biases from the input boundary forcing was discussed and shown in Figure 10. The missing or misrepresented process in tropical and dry regions were discussed in Section 4.1 and 4.2. And last, the observational uncertainty especially in less populated regions such as in polar regions and high mountains were discussed in 4.5 and 4.6.We received additional comments (peer-review-5337105.v1.pdf) and below are our replies.
AC1. It seems that innovation in terms of the methodology is not significant. Please indicatethe innovation of your study at the end of the Introduction, in terms of model development, theory development or case study.
We refined our introduction and included the following statements to highlight the innovation of our methodology: “The objective of this study is to investigate temperature and precipitation biases over the new high resolved domains, as well as, the skill of the model in representing the climatology of all regions in order to identify possible sources of systematic errors of the model inherent to input boundary forcing, observational uncertainty (e.g. [26]), or misrepresented processes (e.g. [27]).” “In this study, the K-T climate types were used to define the regions of analysis using an updated observational dataset.” “These new high resolution simulations will provide additional climate simulations over regions especially with few ensemble members e.g. over Central Asia [16] and Central America [17].” “The simulations will be used to support the growing demands for climate services to provide scientifically sound decisions on climate change adaptation. The coordinated high-resolution simulations could also be used as a basis for further research on climate vulnerability, impacts and adaptation.”AC2. The literature review is not enough. The authors are encouraged to review more papers on climate downscaling methods. Besides, after the literature review, what are their downsides and what is the motivation for you to propose this method?
We have read several papers on climate downscaling methods. Other downscaling approaches would be statistical downscaling, downscaling using a variable grid model such as CCAM or ICON and force them spectrally with the GCM/Renalysis data or using just the SST data as forcing. For instance CCAM was run at global even grid (50 km) forced with bias corrected SSTs. Disadvantages of statistical downscaling on a global scale is the availability of observations, which can be quite limited in many regions. Variable grid models avoid boundary problems but are computationally more expensive than RCMS. In addition, parameters can not be tuned for a specific region. In the case of SST driven simulation there is the advantage to bias correct SSTs before they are used as forcing. Bias correction of the boundary forcing using RCMs is much more complicated because you need to bias correct much more variables. References are:Thevakaran A., McGregor J. L., Katzfey J., Hoffmann P., Suppiah R., Sonnadara D. U. J. (2016): An assessment of CSIRO Conformal Cubic Atmospheric Model simulations over Sri Lanka. Clim. Dynam. 46, 1861–1875. doi: 10.1007/s00382-015-2680-4 Hoffmann P., Katzfey J.J., McGregor J.L., Thatcher M. (2016): Bias and variance correction of sea surface temperatures used for dynamical downscaling. J. Geophys. Res. Atmos. 121, 12877–12890. doi:10.1002/2016JD025383 McGregor, J.L. (2015): Recent developments in variable-resolution global climate modelling. Climatic Change 129: 369. https://doi.org/10.1007/s10584-013-0866-5 However, in this study we focussed on an evaluation method to assess the biases and skill of a regional climate model simulation over several domains using the Koeppen-Trewartha climate types.
AC3. Although the main purpose of the study is stated in this manuscript, the inferences are not drawn clearly from the proposed approach. For example, what could readers learn from the proposed approach? What could be the motivation for the readers to apply the proposed approach to other data sets/study?
AC4. There are many figure issues that need to be modified. For example, in Figure 5 and Figure 7, terrible management and figure size seriously impede the reader to understand what you express. Also, please improve your legend of figures if necessary.
We improved the quality of Figures 5 and 7.AC5. The analysis of your results is abundant, but some of them are lengthy. It is suggested to refine the Discussion section and focus on your core content because your discussion is generally written to some extent.
The study focused on the model skill and sources of biases in ten domains using the Koeppen-Trewartha climate types as analysis regions. In addition, we included previous simulations and other observational datasets, which is crucial in the discussion of sources of biases. We, the authors, tried our best to present relevant information thoroughly in the discussion section.AC6. Please avoid using the first person (e.g., I, We, etc) in formal/academic writing.
Yes, we agree that in the classical form of formal/academic writing, the use of first person personal pronouns (e.g. we) was restricted. However, in contemporary scientific writing, the use of the first person personal pronouns are widely used to engage the readers in reading the journal. Please see Tim Skern’s book on Writing Scientific English. Here’s another interesting article on the use of “we”: http://eloquentscience.com/2011/02/are-first-person-pronouns-acceptable-in-scientific-writing/. In our manuscript, the use of “we” is rather limited and used for clarity.AC7. References should be arranged according to the rules required by the Atmosphere. Please double-check them.
We double-checked the references.Best regards,
Armelle and co-authors
Reviewer 3 Report
This is a well-written manuscript that details the performance REMO in simulating climate types. The authors also provided a lengthy discussion on potential sources of biases. Results could be useful to user community of CORDEX, so I would like to recommend this work for publication subject to a minor revision to address following minor concerns.
1) Overlapping domain (lines 198-200): It is still not clear how the climate classification in the overlapping regions are determined. Overlapping regions in northern South America, Western Africa, and South Asia are significantly large and boundary condition alone won’t be enough to produce consistent climate types between different regional simulations.
2) Model biases are calculated based on pre-determined climate classification from observation (CRU). Given the large biases in some regions, the model may produce a different climate type in some areas. Would it be informative if model simulated classification is compared with observed classification?
3) Model biases are calculated for each climate type. Fig. 5B shows BS in Africa spread quire a lot latitudinally. Even in the same BS, the season in the northern domain is different from the season in the southern domain. So, even though they share the same climate type, they are influenced by different climate. I am wondering whether aggregating the biases in all BS regions may make difficult to examine a potential source of biases.
Author Response
Dear Reviewer,
Thank you very much for your encouraging words! Please find below our replies to incorporate your comments to the improvement of our manuscript:
For data visualization in Figure 3, we overlaid the domains on top of each other to avoid ten figures. However, the biases in the overlapped regions are similar. For the estimation of biases and model skill, we assess the regions of each climate types in each domain individually. For example, the BW in WAS and CAS domains (Figure 5c and 5d) have an area of 10 x 106 km2 and 6 x 106 km2, respectively, based on Table 5. Yes, we agree that it would also add additional information on how the simulated Koppen-Trewartha climate classification is compared with the observed climate classification. However, in this study, as a first step, we focus on the evaluation of the simulations based on the observed climate types. The second step would then be how the climate types are simulated and compared to the future climate regimes, which we are planning to do in future studies. For this paper, as an overview of the biases and skill of the model in simulating ten CORDEX-CORE domains, aggregating the regions into climate types is sufficient especially on dominant climate types such as arid (BW) or continental boreal (Ec) climate. In these climate types, the regions are more clustered. However, in some climate types e.g. BS (semi-arid dry climate), the regions are scattered since they transition from dry climates to subtropical climates. Within this transitional climate types, further analysis is recommended to attribute the sources of biases due to physical processes such as the atmospheric flows.We referred to this issue on Lines 252-257, “A few of the climate masks are shown in Figure 5. Note that the threshold used in determining the climate types are only dependent on the climatological mean of temperature and precipitation. In future studies, the atmospheric flows for each domain should be considered. This is because climate regions are also influenced by different regional flows, such as Aw, BS, and BW (Figure 5). Regions influenced by the same atmospheric flow, as well as overlapping regions (e.g. BW in both WAS and CAS domains in Figures 5c and 5d), should be analyzed further.”
We modified the following lines to incorporate the comments:
“A few of the climate masks are shown in Figure 5. Note that the threshold used in determining the climate types are only dependent on the climatological mean of temperature and precipitation. The regions of analysis are aggregated into similar climate types where the dominant climate types such as arid (BW) or continental boreal (Ec) climate are clustered together. However, in some climate types e.g. BS (semi-arid dry climate), the regions are scattered since they transition from dry climates to subtropical climates. Within this transitional climate types, further analysis is recommended to consider physical processes such as the atmospheric flows. Regions influenced by the same atmospheric flow, as well as overlapping regions (e.g. BW in both WAS and CAS domains in Figures 5c and 5d), should be analyzed further.”
Best regards,
Armelle Remedio (In behalf of the coauthors)
Reviewer 4 Report
The paper presents regional climate model (REMO) simulations over CORDEX domains. The RCM is forced by ERA-interim. The results are compared against the CRU observational
dataset, in terms of temperature (presumably 2m temperature, but this should be specified in the paper) and precipitation. The scores are shown per regional domain, per
saeson, and in particular, per K-T climate zones.
The paper organization is fine, as is the language. Figures are clear and have the appropriate information density. References are abundant and complete.
It might not be clear to the reader how the model spin-up has been performed. The ERA-i data covers the period 1979-2017. The authors state that spin-up time was 30 years,
which leaves 8 years to define the model climate. But the period 1980-2010 is considered for the comparison with the CRU dataset. Please provide more information about the
model simulation design. How is the surface forcing prescribed? Is there any spectral (or other) nudging applied? Please help the readers understand what "the equilibrium"
means in a limited-area model.
My main general concern, however, regards the absent focus on the scores which would support the benefit of the RCM, as claimed in the Introduction. By comparing the
temperature bias of REMO and ERAi against the CRU data, it can be seen that the regional model introduces more, or new bias. This is also clearly seen in the seasonal cycles
of bias (figures 11-19) where the RCM is more often than not the outlier compared to the few observational datasets (or reanalyses). To quantify the added error, the relative
bias maps (figure 3) should be made for REMO compared to ERA-i.
Perhaps the CRM dataset does not enable validations which require time-series analysis? In this case the value of this study is rather limited. Is it possible to include more
climate parameters? The future wind speed trends are increasingly interesting, to plan the development of renewable energy sources, for example.
I would suggest to reduce the range of the color scale in the figures 8 and 9 so that the skill scores are easier to visualize.
Author Response
Dear Reviewer,
Thank you very much for your insights to help improve this manuscript. We revised the methodology to state that in this study, we refer to the temperature at 2 m height as temperature throughout the paper. We added the lines: “The climate variables used in this study were precipitation and the near surface temperature or temperature at 2m height. In this study, we refer to the near surface temperature as temperature throughout the manuscript.”
1. It might not be clear to the reader how the model spin-up has been performed. The ERA-i data covers the period 1979-2017. The authors state that spin-up time was 30 years, which leaves 8 years to define the model climate. But the period 1980-2010 is considered for the comparison with the CRU dataset. Please provide more information about the model simulation design. How is the surface forcing prescribed? Is there any spectral (or other) nudging applied? Please help the readers understand what "the equilibrium" means in a limited-area model.
In each domain, the model was initially spun-up for 30 years from 1979 to 2008 to account for the time the model needs to produce an equilibrium for the soil temperature and soil moisture. These soil fields were then used as the initial soil conditions upon restarting the model from 1979. The lateral boundary conditions were updated every 6 hours. There is no nudging applied in the REMO model. As mentioned in the paper, “The forcing data was prescribed at the lateral boundaries of each domain, which mainly influenced the eight outer grid boxes, with an exponential decrease towards the center of the model domain using a relaxation scheme [55].” We added the following lines: “The model was integrated with a time step of 120 s. For the model to reach an equilibrium state, a spin-up period of thirty years was implemented. In each domain, the model was initially spun-up for 30 years from 1979 to 2008 to account for the time the model needed to produce an equilibrium for the soil temperature and soil moisture. These soil fields were then used as the initial soil conditions upon restarting the model from 1979.” “The forcing data is prescribed at the lateral boundaries of each domain, which mainly influenced the eight outer grid boxes, with an exponential decrease towards the center of the model domain using a relaxation scheme [55]. The lateral boundary conditions are updated every 6 hours.2. My main general concern, however, regards the absent focus on the scores which would support the benefit of the RCM, as claimed in the Introduction. By comparing the temperature bias of REMO and ERAi against the CRU data, it can be seen that the regional model introduces more, or new bias. This is also clearly seen in the seasonal cycles of bias (figures 11-19) where the RCM is more often than not the outlier compared to the few observational datasets (or reanalyses). To quantify the added error, the relative bias maps (figure 3) should be made for REMO compared to ERA-i.
The comparison of the ERA-Interim temperature against CRU was crucial for the discussion on the inherent biases of the model from its driving model. The ERA-Interim temperature is used as one of the initial and boundary conditions to REMO. Figure 10 identified the regions where ERA-Interim already have a bias against CRU. However, the formulation of precipitation in REMO is different with the ERA-Interim reanalysis. The relative bias of precipitation of REMO compared to ERA-Interim would then be a comparison of two models since ERA-Interim has its own scheme for simulating precipitation.3. Perhaps the CRM dataset does not enable validations which require time-series analysis? In this case the value of this study is rather limited. Is it possible to include more climate parameters? The future wind speed trends are increasingly interesting, to plan the development of renewable energy sources, for example.
In this study, we provide the basic climate analysis using precipitation and temperature over ten domains. Other climate parameters are beyond the scope of our study but we indicated in our outlook that analysis of other climate parameters such as wind speeds (atmospheric flows) are worthwhile to investigate in the future.4. I would suggest to reduce the range of the color scale in the figures 8 and 9 so that the skill scores are easier to visualize.
We changed the color scale in Figures 8 and 9.Best regards,
Armelle and co-authors
Round 2
Reviewer 1 Report
The authors have incorportaed my comments and the manuscript has been substantially improved.
Reviewer 4 Report
Thank you for replying to the comments and making the corresponding changes in the manuscript.
I have no further comments.